# Internal uncertainty impacts social information use in risky choice across adolescence

Simon Ciranka [1] ✉ & Wouter van den Bos[1,2]

Adolescents are often thought to be more susceptible to social influence than people in other age groups. This is often explained by altered reward processing or heightened social motivations, such as a need to belong to a group during adolescence. However, uncertainty also makes people more susceptible to social information. While researchers agree that adolescence is a time of great uncertainty, the role of uncertainty in explaining susceptibility to social influence across development remains unclear. Here, we asked 166 participants aged 10–26 to make 144 risky decisions in a lottery experiment, either with or without observing social information and nested within conditions of low and high uncertainty. Modelling susceptibility to social influence as Bayesian updating suggests that despite the same levels of uncertainty between participants, their own internal uncertainty about the utility of choices underwent a negative linear age trend, contributing to age-related differences in susceptibility to social influence across adolescence. Our results suggest that the adolescent development of peer influence is at least in part driven by age differences in the internal uncertainty about how to decide.

During their adolescence, people in Western societies gain more freedom and spend increasingly more time with their peers, away from their caregivers. This freedom comes with opportunities to explore novel behaviours, but for some adolescents, with severe consequences, as evidenced by rising mortality, morbidity, and criminal offence rates in adolescents[1–5]. New opportunities do not necessarily lead to more risky behaviour, but novelty can encourage explorative behaviour that can result in risk-taking[1,6]. Indeed, during adolescence, many people engage in risky behaviours for the first time, such as skipping school[2], experimenting with drugs[7] or having unprotected sex[8]. While these behaviours' consequences can be detrimental, researchers also point out that exploration and risk-taking are essential for adolescent development[9–11]. It motivates adolescents to learn about the world independently ([1,11–14]), and being able to do so seems beneficial for adolescents' mental health[15,16].

Adolescents' behaviours, including their propensity to take risks, are strongly influenced by other people[17–19]. Some suggest that this susceptibility to social influence is driven by adolescents' higher sensitivity to rewards in a social context[20], which makes them particularly prone to take risks[21]. Adolescents may also have an increased sensitivity to social information, which is rooted in attention or motivation to belong to a group[22]. Both, reward sensitivity and social sensitivity impact susceptibility to social information, which refers to behavioural changes resulting from observing social information. However, it is often overlooked that adolescents can also be more susceptible to social influence because they may simply be more uncertain about what to do.

Uncertainty is a state of incomplete knowledge and is part of someone's beliefs about the world, for instance, about the utility that someone expects from their decisions[23]. Gaining experience with similar decisions reduces uncertainty[24]. Uncertainty makes novel information more impactful, because when much is uncertain, there can also be more to learn. Consequently, social information influences people more when they feel more uncertain about how to decide[6,25,26]. Adolescents may often be more uncertain about how to decide than adults or children, since they have less personal life experience to draw from than adults[27], and more agency about their decisions than children. Thus, adolescents' greater uncertainty about the utility of risky decisions could make them more likely to adjust their beliefs and choices in response to social information.

The uncertainty in someone's beliefs is not only shaped by experience, but also by the statistical structure of the environment. Sometimes aspects of the environment can be learned, and through repeated exposure and feedback, individuals can gradually reduce this uncertainty, refine their beliefs about the environment and form better ideas about potential outcomes of their decisions[28]. Often times, however, the environment is hard to predict, making it difficult to form confident expectations and reduce

[1]Center for Adaptive Rationality, Max Planck Institute for Human Development, Berlin, Germany. [2]Department of Psychology, Faculty of Social and Behavioural Sciences, Universtiy of Amsterdam, Amsterdam, The Netherlands. ✉e-mail: ciranka@mpib-berlin.mpg.de

uncertainty about the utility of a decision, regardless of experience. Thus, there is uncertainty that is internal to an individual and is part of someone's beliefs about the environment and the utility of a choice that can be reduced through learning and experience, and there is external uncertainty that is a feature of the unpredictability of the environment itself that cannot be reduced[29].

Adolescents are at a different stage of the process to learn about the world than adults[1,14], and evidence suggests that learning from feedback may be less precise in adolescents[30] compared to adults. This suggests that adolescents' internal uncertainty might be higher than adults', either because they have less experience or because they process feedback differently[31]. Additionally, adolescents might more often be exposed to high levels of external uncertainty in their daily lives, which can, in turn, impact how they learn[32], for example, because they expect a less predictable world than adults. As a result, even with similar experience, adolescents may retain higher levels of internal uncertainty about the expected utility of their decisions—which leads to the social information being more impactful when adolescents make decisions.

In sum, there are three different potential reasons why adolescents are more uncertain. First, they have less experience. Second, they may learn from feedback differently. Third, they may encounter more external uncertainty in their environment in general.

Recent research suggested that adolescents also treat uncertainty differently from people in other age groups. When there is no social information, it sometimes seems as if adolescents are more "tolerant of uncertainty" than people in other age groups. This means, adolescents do not avoid uncertain choice options to the same extent as adults do[33,34]. At the same time, experimental research showed that adolescents are indeed more influenced by others, precisely when decision outcomes are more uncertain[35]. Thus, adolescents' choices under uncertainty seem characterized by both, a greater openness to choose uncertain options, and also a greater susceptibility to social input when navigating uncertain choice options. Internal uncertainty about the utility of a choice can explain both, it makes choosing risky more likely compared to adults who have a more certain tendency to avoid risk[36], and makes social information more informative.

Taken together, developmental differences in susceptibility to social influence across adolescence can have different reasons. It could be that social information makes adolescents crave rewards more. It could also be that adolescents have a higher general sensitivity to social information and are more likely to conform to what others do or they think others want[37]. Finally, adolescents' internal uncertainty could be higher, for instance, they could be more uncertain than adults about the utility of their choices, making them more susceptible to social influence. Here, we formulate a cognitive model of social influence where participants, after seeing social information, update their beliefs about the utility of a choice proportionally to the internal uncertainty they experience about that choice. On October 17th, 2018 we preregistered (https://osf.io/nsy69) the following hypotheses:

**H1:** Adolescents make more risky decisions than adults and children when external uncertainty is high, not when it is low, because they are thought to be tolerant to uncertainty[33,34], which means they take more risks than adults when external uncertainty is high.

**H2:** Adolescents are more influenced by social information than adults or children.

**H3:** Greater uncertainty leads to greater susceptibility to social information.

**H4:** If H1 is true, then adolescents will already take more risks than adults or children when external uncertainty is high. This, in turn, means that adolescents' susceptibility to (risky) social influence should be less affected than the susceptibility to (risky) social influence of adults or children by our experimental manipulations of uncertainty.

**H5:** A model, sensitive to safe and risky behaviour of an advisor, describes choices better than alternative models.

Note that we reformulated these hypotheses compared to the preregistration and only report the confirming results for H5 in the supplement, and elaborate on the hypothesis at length elsewhere[36].

## Methods

### Participants
We report the data from 76 female and 90 male ($n = 166$, sex self-reported) German participants from the Berlin Dahlem area (aged 10–26, $m = 15.82$) who were invited to our laboratory (see Fig. S1 for age distribution, puberty status and sex). Participants received a payment of 10€ for their participation and could get a bonus based on their decisions in the experiment. The experiment was programmed using jsPsych[38] and administered in a regular web browser on a computer in the lab. On the same occasion, participants also completed other social decision-making tasks reported elsewhere[39], the digit-span test for working memory capacity[40] and the CFT-20R[41] test for general cognitive ability. Participants provided their informed consent to these procedures, and the local ethics committee approved the study (protocol number A-2018-22).

### Task and procedure
Participants were asked to make a series of risky decisions under high and low external uncertainty, with and without additional social information about how someone else chose between the same alternatives. The task consisted of 144 trials, in each of which participants saw two marble jars. The jars differed in the proportion of red and blue marbles in them. We instructed participants to choose which jar to draw a random marble from in each trial. Participants would win points if they drew a blue marble from the jar.

There was a safe and a risky jar in each trial. The jars differed in the probability of drawing a blue marble and in how many points a blue marble would promise. The safe jar contained only blue marbles, and choosing the safe jar would always result in 5 points. The risky jar contained both blue and red marbles. A risky jar promised more points (8, 20, or 50 points), which, however, were awarded only with some probability (probabilities were 0.125, 0.25, 0.375, 0.5, 0.625, or 0.75). The probabilities corresponded to the proportion of blue marbles in the jar. Every combination of points and probabilities was shown eight times in randomised order. In choosing the risky jar, participants risked gaining nothing for the prospect of gaining more points. The points they accumulated were translated into a real monetary bonus at the end of the experiment. The bonus payment in € was determined by multiplying the points collected by 0.008.

### Experimental manipulation of external uncertainty
We manipulated external uncertainty by asking participants to make decisions from description, where external uncertainty is low, or decisions from experience, where external uncertainty is high[42]. When participants made decisions from description (Fig. 1a), the winning probability was shown to them unambiguously. Participants saw 100 red and blue marbles in the jar, with the number of blue marbles indicating the probability of winning.

External uncertainty was higher in decisions from experience (Fig. 1b), where participants learned about outcome probabilities by observing nine samples from the jar (Fig. 1b; left). These nine samples of red and blue marbles were shown in five $3 \times 3$ grids with an interstimulus interval of 250 ms. There were also grey circles that served as placeholders; this was irrelevant to the current task but ensured fMRI compatibility in a planned follow-up study. We predetermined the sequences of red and blue marbles so that for each condition, every participant saw the same number of red and blue marbles. The samples were chosen to be as representative of the generative probability as possible. We then asked participants to estimate the proportion of blue and red marbles in the jar using a slider (Fig. 1b; middle) before making a decision from experience (Fig. 1b; right).

### Experimental manipulation of social information
In the second half of the experiment, participants saw social information about how someone else chose previously (Fig. 1a, b, bottom) in the same lottery. This was done by highlighting the others' choice with a speech bubble around the option that was chosen by the other. Participants knew nothing about the identity of the person whose decisions we showed, only

**Fig. 1 | The marble task.** Participants choose between a risky or a safe jar after learning about possible outcomes and their probabilities either **a** from description, where external uncertainty is low or **b** from experience, where external uncertainty is high. Participants see social information about the choices of another participant (bottom) or not (top) in half of the trials each. We retrieved this social information from the decisions of participants in a previous experiment using the same probability-value combinations[43].

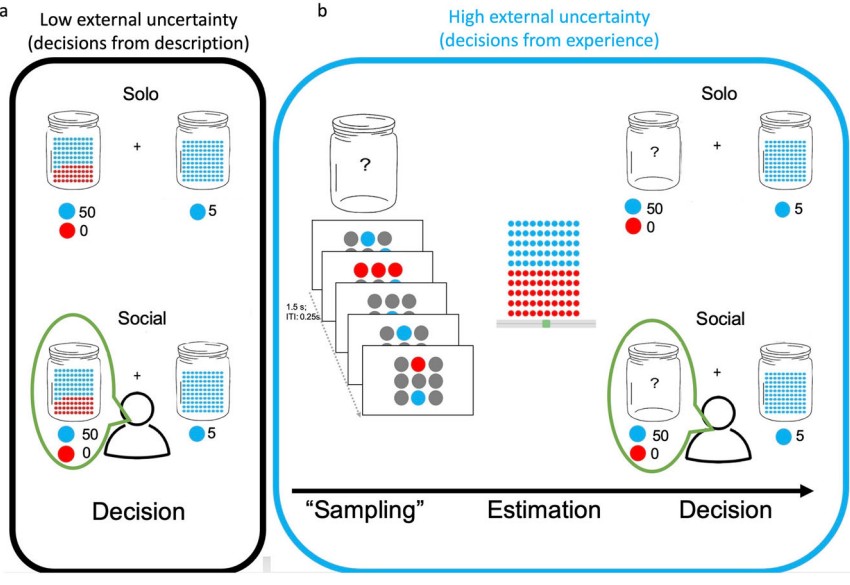

that we tried to find someone matched to their risk preference. We did that by finding a participant of a previous experiment[43] who, on average, made 15 more risky decisions (20%) than the participants in the first half of the experiment. We chose this criterion to increase our ability to detect age differences in an increase in risky choices, but still show conceivably human choices. When it was not possible to find someone who made 15 more risky decisions, we relaxed this criterion in integer steps until we found someone.

To ensure that even the youngest participants understood the task, an experimenter read out detailed instructions, asked for a comprehension check, and moderated a short practice round before participants started the task. Furthermore, the instructions included three built-in comprehension checks that participants had to pass before proceeding with the experiment (see supplementary material for screenshots of the instructions).

**Statistical analyses.** All statistical analyses were done in stan[44] and the brms package[45] with reliance on software provided by the tidyverse ecosystem[46] for the programming language R[47]. For statistical inference, we relied on 95% credible intervals of the posterior and Bayes factors. We report Bayes factors (BF$_{10}$) as the Savage Dickey density ratio between priors (Cauchy priors $\theta = 0$ and $\sigma = 0.2$) and posterior estimates of the regression weights. In general, Bayes factors that are larger than 3 are considered substantial evidence that the predictor has an impact on the dependent variable (the alternative hypothesis). Bayes factors smaller than 0.3 are considered substantial evidence in favour of the null hypothesis that there is no impact of the predictor. Bayes factors between 0.3 and 3 are considered inconclusive. All continuous variables were standardised to have a mean of 0 and a standard deviation of 0.5. Regressions were run with 10,000 iterations of four Markov chains. The convergence of the Markov chains was assessed by consulting the Gelman-Rubin statistic, $\hat{R}$, which was 1 for all reported regressions, indicating convergence.

**Regression model**

We analysed the preregistered hypothesis with a generalised linear mixed model, predicting each decision using a logit link function to a Bernoulli likelihood function. The assumptions of the generalised model were not formally tested. The regressions were specified as follows: When participants chose risky, this was coded as "1", a safe choice was coded as "0". For the regressors, low external uncertainty (decisions from description) were coded as factors with level "0" and high external uncertainty (decisions from experience) as "1". Social information was coded as factors with "-1" for no social information, "0" for safe social information, and "1" for risky social

information. This coding makes "no social information" the reference category, allowing us to compare the impact of social information favouring either the risky or safe option. Any regression term involving social information (whether it was risky or safe) or its interactions with another factor evaluates the difference between choices made without social information and those made with either risky or safe social information. The expected value of each decision was entered as a continuous predictor. Linear and quadratic polynomials of our participants' age were continuous predictors. Finally, we specified interactions with age and quadratic age as social information, as well as whether participants made decisions from description or experience. Note that for the sake of simplicity, we often refer to groups (children, adolescents and adults) in the text but treat age as continuous in our analyses since any clear-cut age-group boundary of adolescence is hard to justify (Table S1 shows, however, that analysis with age-bins offers the same conclusion). By including regressors of participants' test scores for working memory capacity[40] and fluid intelligence[41], we attempted to control for cognitive factors which might otherwise be a confound. Finally, we included random intercepts per participant.

**Deviation from the preregistered regression model**

We preregistered to code social information as "1" when there was social information and "0" when there was no social information. However, this was not ideal, as it would have made us unable to detect differences between the influence of risky and safe social information, and these opposing social signals' effects on our participants choices might even cancel each other out on average. We also note that while we preregistered a statistical model, we failed to preregister inference criteria and how the results of the model would lead us to accept or reject the hypotheses. We here interpret the posterior regression coefficients and Bayes factors according to general conventions in the literature and added marginal effects plots in the supplement to aid their interpretation.

**Modelling social influence in risky choice as internal-uncertainty updating.** With cognitive modelling, we quantified developmental differences in participants' internal uncertainty and how it impacts their susceptibility to social influence. In the model (Fig. 2), people update their beliefs about the expected utility of a jar when they see advice favouring that jar. Following Bayesian principles, the model updates beliefs about the utility more in the face of social information when participants have more internal uncertainty about the utility.

The model has four components, each of which specifies a simpler model of task behaviours. In model recovery analyses (see Fig. S7), we

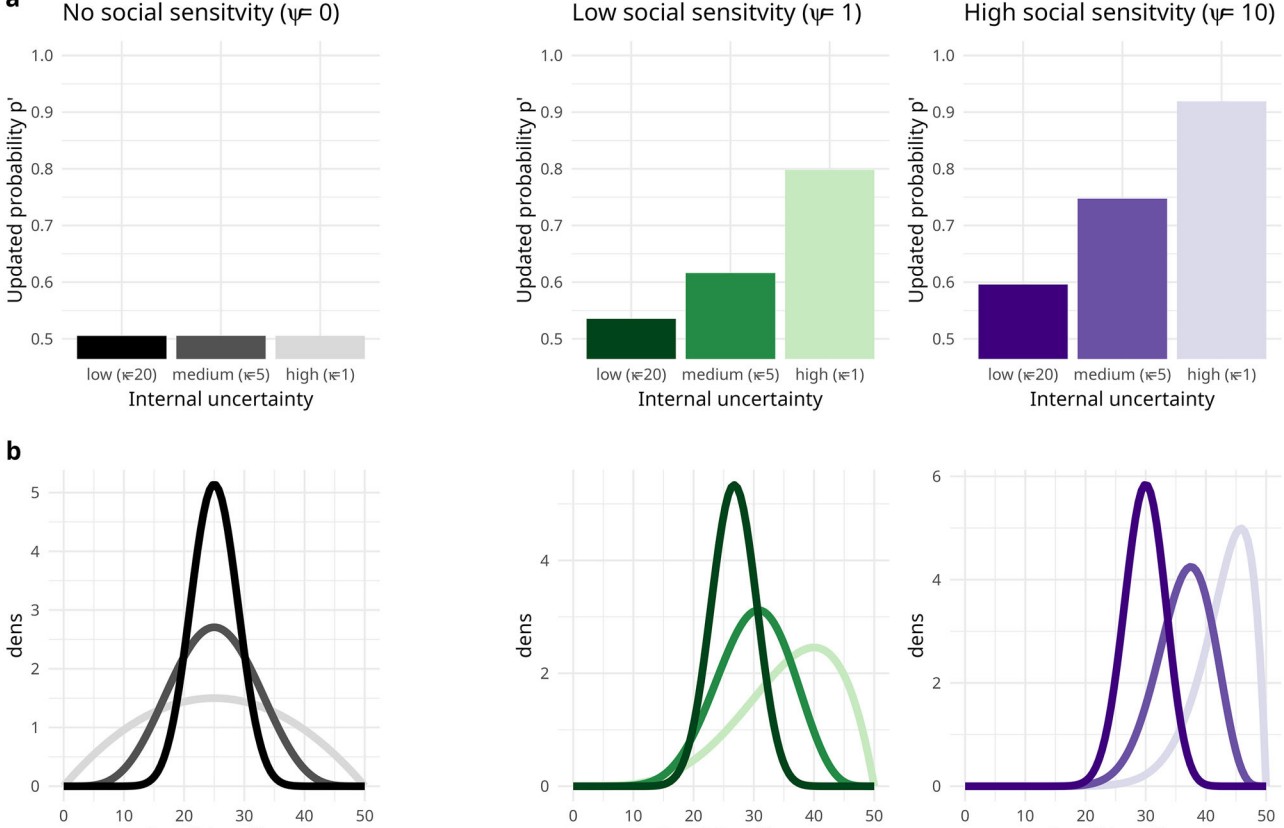

**Fig. 2 | Illustration of the cognitive model.** The model formalises susceptibility to social information in our experiment as Bayesian updating. Susceptibility to social information depends on participants' internal uncertainty (x-axis in **a**; opacity) and their social sensitivity (panels). The example shows participants' subjective representation of the outcome probabilities of a lottery with an objective probability of 0.5 of drawing a blue marble (of winning) under different values for internal uncertainty ($\kappa$) and social sensitivity ($\psi$). **a** The panel denotes the effect of parameter $\kappa$, x-axes; opacity, which models participants' internal uncertainty; and the parameter $\psi$ (colours; panels), which models an orthogonal social sensitivity. The leftmost panel

(Greyscales) describes beliefs without social information (Eq. 3) or of participants with no social sensitivity. Bars show how more internal uncertainty (x-axis) leads to a larger shift in participants' beliefs about the probability (y-axis). In this case, p shifts towards believing it is more likely to draw a blue marble after seeing social information favouring the risky option. **b** Shows the possible expected utilities of the advised jar (x-axis) change after seeing social information in favour of it, and this change is stronger when prior uncertainty (colour saturation; x-axis in **a**) about the probability of winning is higher.

demonstrate that each element contributes to the full model's ability to explain behaviour in the experiment.

The simplest model assumes that social information has no impact on choices and that only individual differences in participants' reward sensitivity determine differences in their choices. A more complex model assumes that social information impacts our participants' decisions, akin to an ideal Bayesian observer, where social information has a more significant impact when people are more uncertain, and where differences in internal uncertainty correspond to differences in susceptibility to social information. Finally, the most complex model assumes that people also differ in social sensitivity, which can make social information more or less impactful than it should be from a Bayesian perspective. We introduce these components step by step in the following.

First, people transform outcome values of the lotteries into a utility with a power function, where exponent $\rho$ models the curvature of the function. $\rho < 1$ describes risk-aversion and $\rho > 1$ risk-seeking attitudes:

$$U = V^\rho \tag{1}$$

Second, participants are internally uncertain about the utility they can expect if they take risks. We model this internal uncertainty using a Beta distribution with the lottery's outcome probability $p$ as the mean and a strictly positive rate parameter, $\kappa$, that can take different values in decisions

from description and experience (Fig. 2a and y-axis Fig. 2b):

$$\phi \sim beta(p, \kappa). \tag{2}$$

$\kappa$ models individual differences in our participants' internal uncertainty, with lower values implying more internal uncertainty. For example, $\kappa = 2$ implies that participants act like they saw two marbles to inform their probability estimate; participants with $\kappa = 20$ act as if they saw 20 marbles to inform their estimate and thus act as if their internal uncertainty was lower.

Third, advice skews this distribution and results in a riskier or safer decision policy than when participants decide alone. To model this, we re-parameterise the beta distribution to $\alpha = p * \kappa$ and $\beta = (1 - p) * \kappa$, with $\alpha$ and $\beta$ representing the number of successes and failures in a binomial trial:

$$p = \frac{\alpha}{\alpha + \beta}, \tag{3}$$

where $\alpha$ now relates to the number of blue marbles and $\beta$ the number of red marbles. This allows for a straightforward implementation of Bayesian updating in social trials by adding a value, $SI$ to $\alpha$ or $\beta$:

$$p' = \frac{\alpha + SI_{risk}}{(\alpha + SI_{risk}) + (\beta + SI_{safe})}. \tag{4}$$

$SI_{risk}$ or $SI_{safe}$ equal 1 when social information favoured the risky or safe choice, respectively, and 0 otherwise. This is an ideal Bayesian observer model where social information favouring the risky choice suggests that there are relatively more blue marbles (red marbles for safe advice) in the jar than the individual initially thought.

In Eq. 4, SI is a dummy variable that codes whether social information favoured the risky or safe option, which implements Baysian updating. But participants may not be ideal Bayesian observers, but also differ in their sensitivity to social information, irrespective of their internal uncertainty, meaning, for instance, they differ in whether they treat social information as equal to two or 20 marbles.

We model these differences in an orthogonal sensitivity to social information with the parameter $\psi$ (colours in Fig. 2). This version of the model suggests that people differ in ther social motivations or beliefs about others, leading to individual differences in suceptibility to social information beyond participants' internal uncertainty. A $\psi$ of 0 means that participants ignored social information (Fig. 2, grey). By invoking separate parameters for $\psi$ (Eq. 5) depending on whether social information was safe or risky, this model updates the probabilities with differential sensitivity to safe vs risky social information that is often reported in the literature[36,48,49]:

$$p' = \frac{\alpha + SI_{risk} * \psi_{risk}}{\left(\alpha + SI_{risk} * \psi_{risk}\right) + \left(\beta + SI_{safe} * \psi_{safe}\right)}. \tag{5}$$

By virtue of Bayes' theorem, the difference between $p$ (Eq. 3) and $p'$ (Eq. 5) is high when internal uncertainty is high and low when internal uncertainty is low. This means that if participants only saw one red and one blue marble ($\kappa = 1 + 1$), social information favouring one jar would greatly impact their beliefs, much more than when they saw ten red and ten blue marbles ($\kappa = 10 + 10$), even when the mean of the prior belief was the same before seeing social information (see Fig. 2b). With these updated probabilities, the expected utility of choosing the risky jar is computed as:

$$EU_{risk} = p' * U. \tag{6}$$

Per convention, a risky choice is predicted by the difference between the expected utilities of the risky and safe jars with a sigmoid choice function that has a temperature parameter $\tau$. A higher temperature parameter implies more randomness in our participants' choices, akin to the increased randomness of molecule movements when the temperature of a system increases:

$$cp = \frac{1}{1 + e^{-\frac{EU_{risk} - EU_{safe}}{\tau}}}. \tag{7}$$

We compared the models' predictive accuracy using approximate leave-one-trial-out cross-validation and the resulting leave one out information criterion (looic—a fully Bayesian version of classical model fit indices such as AIC and BIC).

Models were implemented in the probabilistic programming language stan[44] and fitted with stans' Hamiltonian Monte Carlo algorithm. Every model was fitted using four Markov chains of 6000 iterations; the first 2000 iterations of every chain were discarded as warmup. We implemented the model hierarchically; with age groups that we preregistered in our recruitment strategy as hyperpriors (see Fig. S3 for a graphical model and priors). To understand the relationship between age and the parameters of the most accurate cognitive model (see next section), we estimate a multivariate Bayesian linear regression and predict parameters of the cognitive model with linear and quadratic polynomials of our participants' age. We preregistered the computational model as an exploratory analysis and did not specify particular hypotheses for this analysis.

### Reporting summary

Further information on research design is available in the Nature Portfolio Reporting Summary linked to this article.

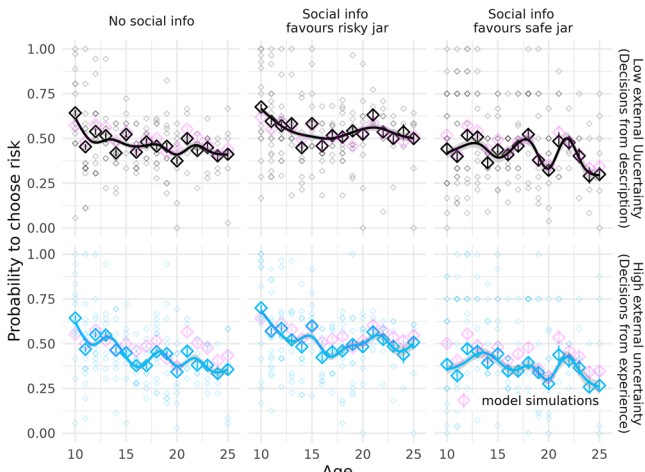

**Fig. 3 | Behavioural results ($n = 166$ participants).** Proportions of risky decisions in the marble task (*y*-axis) by age (*x*-axis) under uncertainty and risk. Solid diamonds show means, and error bars show bootstrapped 95% confidence intervals. Transparent diamonds show individual participants' means. The solid lines show the fit of a general additive model predicting risky choice proportions with age. Column panels depict whether social information was safe, not present or risky. Pink shapes denote posterior predictions from the Bayesian utility model described in Eqs. 5–7 and Fig. 2.

## Results

In the experiment, we asked participants aged 10–26 to make decisions from description and from experience without and with accompanied social information. First, we will report the main effects of the experimental manipulations, which also serve as a sanity check, before we turn to our preregistered hypotheses, which we tested by examining the interaction effects of the experimental manipulations with age (Fig. 3). Table 1 denotes the odds ratios, that is the change in probability of choosing the risky option that can be attributed to changes in the predictor in the rows.

To aid the interpretation of the regressions, we provide plots of the marginal effects of all regression weights in question (Fig. 4). Finally, we will examine age differences in the cognitive model parameters.

### Behavioural analyses

Overall, participants chose the risky jar more often when the expected value of the risky jar was higher (b_EV = 4.11, 95% CI = [3.98, 4.24], BF$_{10}$ > 100; Fig. 4b), and participants chose the risk jar less often when making decisions from experience compared to description (b_experience = −0.14, 95% CI = [−0.24, −0.05], BF$_{10}$ > 100; Fig. 4c). Participants chose the risky jar more often when social information favoured the risky jar (b_socialrisk = 0.27, 95% CI = [0.16, 0.37], BF$_{10}$ > 100; Fig. 4d). However, unexpectedly, on average, participants did not choose safe more often when social information favoured the safe jar (b_socialsafe = 0.05, CI = [−0.09–0.18], BF$_{10}$ = 0.37; Fig. 4d).

In line with previous studies, risky choice declined with age on average across conditions (b_age = −0.52, CI = [−1.01, −0.05] BF$_{10}$ = 8.62; Fig. 4f), and there was no quadratic age trend (b_age$^2$ = −0.17, CI = [−0.61, 0.31], BF$_{10}$ = 0.60; Fig. 4j) that would be indicative of an adolescent-specific component in the propensity to choose riskily. Finally, risky choice was not related to individual differences in working memory (b_digitspan = 0.02, CI = [−0.31, 0.38], BF$_{10}$ = 0.58) or fluid intelligence (b_cft = -0.08, CI = [−0.44, 0.24], BF$_{10}$ = 0.63).

### Behavioural analyses of preregistered interaction effects

The first preregistered hypothesis (H1) was that there would be an adolescent peak in decisions from experience but not from description. A positive interaction between quadratic age and the high external uncertainty condition requiring decision from experience as opposed to description

## Table 1 | Results of the regression model testing preregistered hypotheses

| Predictors | Propensity to choose risk | | |
|---|---|---|---|
| | Odds ratios | 95% CI | Bayes factor |
| Intercept | 1.10 | 0.89–1.37 | 0.05 |
| Expected value | **61.17** | **53.74–69.54** | **>100** |
| Other safe | 1.05 | 0.92–1.20 | 0.371 |
| Other risky | **1.31** | **1.18–1.44** | **>100** |
| Uncertainty | **0.87** | **0.79–0.95** | **22.32** |
| Linear age | **0.59** | **0.35–0.95** | **8.62** |
| Quadratic age | 0.94 | 0.65–1.27 | 0.60 |
| Fluid intelligence (cft score) | 0.94 | 0.64–1.28 | 0.63 |
| Memory capacity (digit span score) | 1.02 | 0.73–1.46 | 0.58 |
| Other safe × uncertainty | **0.77** | **0.63–0.94** | **15.46** |
| Other risky × uncertainty | 1.03 | 0.86–1.19 | 0.30 |
| Other safe × linear age | **1.41** | **1.07–1.89** | **11.90** |
| Other risky × linear age | 1.01 | 0.86–1.19 | 0.32 |
| Other safe × quadratic age | **1.42** | **1.09–1.85** | **18.00** |
| Other risky × quadratic age | 0.92 | 0.77–1.09 | 0.50 |
| Uncertainty × linear age | **0.74** | **0.61–0.87** | **>100** |
| Uncertainty × quadratic age | 0.94 | 0.81–1.10 | 0.36 |
| Other safe × uncertainty × linear age | 1.36 | 0.96–2.10 | 2.66 |
| Other risky × uncertainty × linear age | 1.09 | 0.88–1.41 | 0.61 |
| Other safe × uncertainty × quadratic age | 1.11 | 0.84–1.58 | 0.72 |
| Other risky × uncertainty × quadratic age | 0.94 | 0.74–1.16 | 0.49 |
| **Random Effects** | | | |
| $\sigma^2$ | | 3.29 | |
| $\tau_{00\ participant}$ | | 1.75 | |
| ICC | | 0.35 | |
| $N_{participant}$ | | 160 | |
| Observations | | 23025 | |
| Marginal $R^2$/Conditional $R^2$ | | 0.360/0.447 | |

We predicted risky decisions on each trial using predictors in the first row and a random intercept for participants. The second row shows the odds ratio, which is the exponential of the regression coefficient. An odds ratio greater than 1 indicates an increase in odds of a risky choice along a change of the predictor, while an odds ratio less than 1 indicates a decrease. The third row denotes the 95% credible interval of the posterior odds ratio. The fourth row shows Bayes factors, comparing the posterior with a Cauchy prior with location 0 and dispersion 0.2. Bold values denote regression estimates with Bayes factors larger than 3 and thus are considered "substantial" by common convention.

would align with that hypothesis. However, this interaction was not credible (b_age$^2$ × uncertainty = −0.06, CI = [−0.21, 0.09], BF$_{10}$ = 0.36; Fig. 4k). The interaction was negative and credible for the linear age term (b_age × uncertainty = −0.31, CI = [−0.50, −0.13], BF$_{10}$ > 100; Fig. 4g), suggesting that participants' propensity to choose risky was less strong in older participants than younger ones when external uncertainty was higher and required making decisions from experience.

The second preregistered hypothesis (H2) stated that the effect of social information would be generally stronger in adolescents than in other participants, which we tested with the interaction between quadratic age and the variable that coded whether social information favoured the risky or the safe jar. Contrary to our expectations, observing social information favouring the risky jar similarly affected participants of all ages (b_risky × age = −0.01, CI = [−0.16, 0.18], BF$_{10}$ = 0.32, Fig. 4h; b_risky × age$^2$ = −0.08, CI = [−0.30, 0.13], BF$_{10}$ = 0.54; Fig. 4l). There was a linear (b_safe × age = 0.35, CI = [0.7, 0.64], BF$_{10}$ = 11.90, Fig. 4h) and a quadratic (b_safe × age$^2$ = 0.35, CI = [0.09, 0.61], BF$_{10}$ = 18.00; Fig. 4i) age trend in

participants' susceptibility to social influence after seeing safe social information. These linear and quadratic effects taken together suggest an initially strong and then plateauing age-related decline of susceptibility to social influence after seeing safe social information, which, judging from visually inspecting the marginal predictions (Fig. S2k), might even be reversed in the oldest participants. The oldest participants seem more (and not less) likely to choose risky when seeing safe social information, compared to when seeing no social information.

Our third hypothesis (H3) was that people would be more susceptible to social information when there is greater external uncertainty, such as when they make decisions from experience rather than description. We tested this with the interaction between social information favouring the risky and the safe jar and whether participants made decisions from description or experience. There was no uncertainty dependent change to the difference between choices without social information and social information that favoured the risky jar (b_risky × uncertainty = 0.03, CI = [−0.10, 0.17], BF$_{10}$ = 0.30, Fig. 4e). However, when social information favoured the safe jar, participants were less likely to choose risky than when there was no social information; in particular when external uncertainty was higher in decisions from experience (b_safe × uncertainty = -0.26, CI = [−0.46, −0.07], BF$_{10}$ = 15.46; Fig. 4e).

In H4, we hypothesised that susceptibility to social influence would differ less between decisions from description and experience in adolescents. This corresponds to a negative three-way interaction between social information favouring the risky or safe jar, whether participants made decisions for description or experience, and quadratic age. This interaction was not credible either for social information favouring the risky (b_risky × uncertainty × age$^2$ = −0.07 CI = [−0.30, 0.15], BF$_{10}$ = 0.49; Fig. 4m), nor the safe jar (b_safe × uncertainty × age$^2$ = 0.11, CI = [−0.17, 0.46], BF$_{10}$ = 0.72; Fig. 4m). There also were no credible three way interactions with risky (b_risky × uncertainty × age = 0.09 CI = [−0.13, 0.35], BF$_{10}$ = 0.61; Fig. 4i) or safe social information (b_safe × uncertainty × age = 0.32 CI = [−0.04, 0.74], BF$_{10}$ = 2.06; Fig. 4i). This means that there was no developmental difference in the interaction between different degrees of external uncertainty and being influenced by risky or safe social information.

Summarising the age trends of our behavioural results indicates that older participants choose more cautiously than younger ones when there is high external uncertainty requiring decisions from experience, with no adolescent peak in risky choices. Overall, younger participants seemed more susceptible to social influence when social information favoured the safe, but not the risky jar, a difference that remained statistically similar no matter if external uncertainty was low or high.

### Model analyses

Model comparison using leave-one-out cross-validation indicates that the model specified by Eqs. 5–7, in which the impact of social information on our participants' decisions depends on their internal state of uncertainty in a quasi-Bayesian fashion, accounted best for our participants' behaviour in comparison to their simpler counterparts (Fig. S3).

We were most interested in how age differences in participants' susceptibility to social influence were related to our participants' internal uncertainty ($\kappa$) and social sensitivity ($\psi$), which the model separates. For this, we first confirmed that the model reproduces our participants' decisions (Fig. 3; pink shapes) and that model parameters can be reliably identified by fitting the model to simulated data for which parameter values are known (Figs. S4 and S5).

Inspecting parameter $\kappa$ revealed that there were no age differences in participants' internal uncertainty about how to choose when making decisions from description (b_age = 0.01, CI = [−0.08, 0.09], BF$_{10}$ = 0.17; b_age$^2$ = 0.05, CI = [−0.03, 1.14], BF$_{10}$ = 0.36 Fig. 5b: $\kappa$ black). When making decisions from experience, however, the linear age effect was substantial (b_age = 0.28, CI = [0.12, 0.44], BF$_{10}$ > 100; b_age$^2$ = 0.3, CI = [−0.11, 0.18], BF$_{10}$ = 0.31, Fig. 5b: $\kappa$, blue), suggesting when external uncertainty was higher, older participants had less internal uncertainty

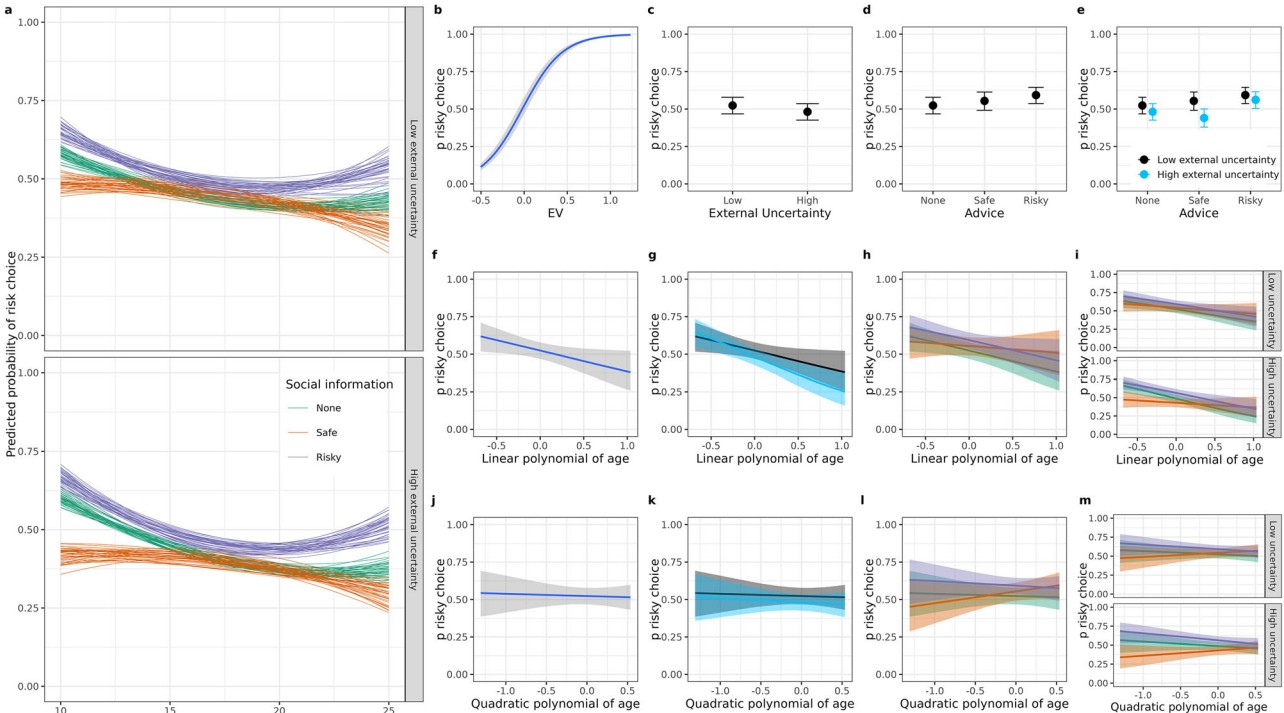

**Fig. 4 | Regression results ($n = 166$ participants). a** Shows regression model predicted age trends in risky choice when there was no social information (green), social information favouring the safe (orange) or risky (purple) marble jar. Each line refers to predictions of one sample from the posteriors of the regression model.
**b–m** Denote marginal effects and depict how the probability of choosing risky ($y$-axis) differs with changes in the independent variables ($x$-axis), holding other variables constant. Two-way interactions are illustrated with colours. Interactions with age and social information are represented with green, orange and purple for none, safe and risky social information (**h, l**). External uncertainty is represented with blue and black when it is high and low, respectively. Three-way interactions of age, uncertainty and social information (**i, m**) always show uncertainty conditions in panels and social information in colour. The second row shows linear and the third row shows quadratic polynomials of our participant's age. The quadratic marginal effect represents both, the youngest and the oldest participants on the left side of the $x$-axis, whereas the rightmost side of the $x$-axis represents 18-year-olds. It is important to note that these rows should not be interpreted in isolation. The polynomials are needed to model non-linear age differences and are combined in the predictions shown in a). Error bars (**c, d**) and ribbons (**a, f–m**) show the effects that lie within the 95% credible interval of the regression weight.

about what to choose relative to younger participants, with no credible non-linear age trend.

In line with previous research[36], there were credible negative linear and quadratic age trends for social sensitivity to safe (b_age = -0.15, CI = [−0.19, −0.11], $BF_{10} > 100$; b_age$^2$ = −0.12, CI = [−0.16, −0.08], $BF_{10} > 100$ Fig. 5c:, $\psi_{safe}$) and to risky (b_age = −0.11, CI = [−0.17, −0.06], $BF_{10} > 100$; b_age$^2$ = −0.8, CI = [−0.13, −0.03], $BF_{10} = 11.15$ Fig. 5c:, $\psi_{risky}$) social information.

Also in line with previous research, the temperature parameter $\tau$ was smaller in older participants (b_age = −0.28, CI = [−0.40, 0.16], $BF_{10} > 100$ Fig. 5e:, $\tau$), with a quadratic age component indicative of a steeper age-related decrease of temperature in younger participants (b_age$^2$ = −0.16, CI = [−0.27, −0.04], $BF_{10} > 100$ Fig. 5e: $\tau$). In other words, older participants were less likely to choose risky when expected values of the risky jar are low, but more likely to choose risky when expected values were high. Adding the respective interaction post-hoc in the regression model reported above confirms this finding (b_EV × ge =2.03, CI = [0.18, 2.25], $BF_{10} > 100$; Fig. S8; other regression weights were unchanged by including this interaction). A positive interaction between expected value and age could mean that older people are more sensitive to the reward promised by the lottery's expected value. Notably, modelling suggests that this expected value-age interaction is likely not driven by age differences in reward sensitivity. The model parameter $\rho$ is a more direct measure of reward sensitivity and was highest in our youngest participants (b_age = −0.01, CI = [−0.02, 0.00], $BF_{10} = 1.60$; b_age$^2$ = −0.01, CI = [−0.01, −0.00], $BF_{10} = 0.04$; Fig. 5d: $\rho$) in our data just as in previous research, although our age effects were not substantial.

Finally, to better quantify the joint impact of our participants' internal uncertainty and social sensitivity on their susceptibility to social influence,

we computed how strongly their beliefs differed before and after seeing social information. We measured this difference with the participant-level average change in the probability between the prior (p) and posterior (p') probability of drawing a blue marble before and after seeing social information (Fig. 4a). A value of 0.2 means that participants acted as if the advised jar was 20% more likely to result in a desired outcome than when not seeing social information.

Examining developmental differences in this susceptibility to social influence, we computed a regression with linear and quadratic age decisions from description coded as "0" and decisions from experience coded as "1" as predictors of the absolute difference between p and p'. In line with H2, susceptibility to social influence was bigger when external uncertainty was higher (b_experience = 0.08, CI = [0.02,0.14], $BF_{10} = 5.65$). Susceptibility to social influence was smaller in older participants (b_age = −0.20, CI = [−0.34, −0.06], $BF_{10} = 15.91$). There was no credible quadratic age component (b_age$^2$ = −0.12, CI = [−0.26 0.01], $BF_{10} = 1.60$). There were also no age interactions (b_age × experience = −0.85, CI = [−0.19, 0.03], $BF_{10} = 0.51$; b_age$^2$ × experience = -0.01, CI = [−0.12, 0.10], $BF_{10} = 0.21$), with whether participants made decisions from description or experience.

In sum, susceptibility to social influence was more pronounced in younger participants than in adults for two reasons. First, younger participants were more internally uncertain than older ones, which made them additionally susceptible to following social information. Second, social sensitivity was smaller in older participants. Adolescent peaks or minima in any parameter or behavioural measure were notably absent in our data. Our results instead suggest that, on top of decreasing social sensitivity, internal uncertainty decreases steadily with age, and people become less unsure about the utility of their choices. Model comparison, parameter inference,

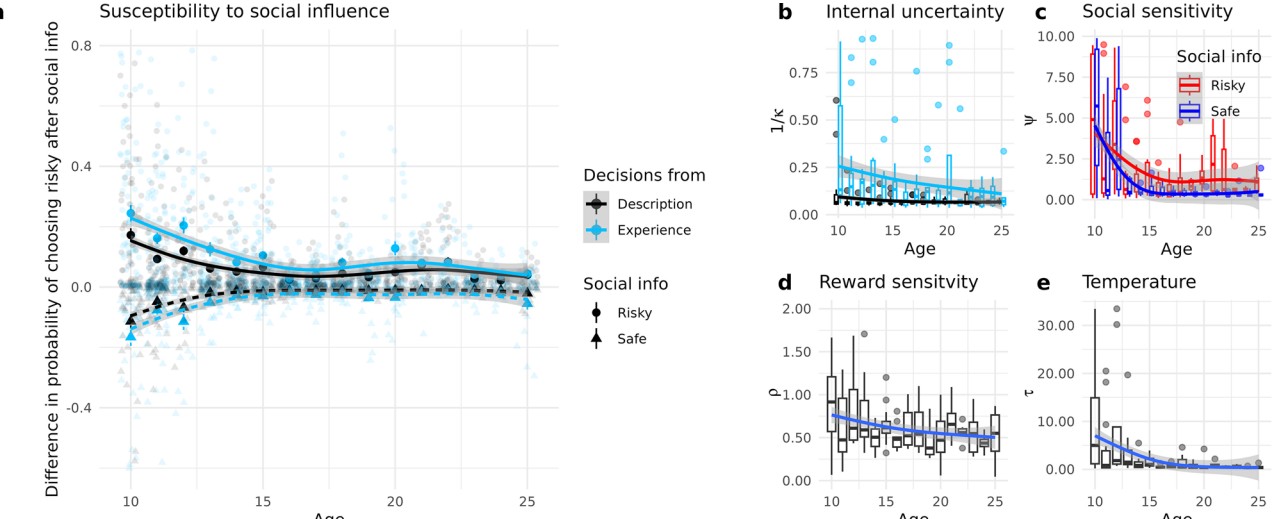

**Fig. 5 | Age trends in the social influence model ($n = 166$ participants).**
**a** Susceptibility to social influence. Difference between the beta distribution mean for $p$ and $p'$ as a measure of social impact ($y$-axis) by age ($x$-axis). Black lines and shapes represent decisions from description, and blue lines and shapes represent decisions from experience. Round shapes and solid lines show trials with safe, triangles and dotted lines show trials with risky social information. Positive values indicate that participants were more likely to choose a risky option, and negative values indicate that participants were more likely to choose a safe option after seeing social information. Shapes show the average in every age bin, the error bars denote

the bootstrapped 95% confidence interval, and the lines show a linear fit. **b–e** Model parameters. Panels depict the population statistics of posterior means of model parameter estimates for each individual ($y$-axis) by age ($x$-axis). Boxplots show the interquartile range, whiskers the 95% confidence interval, dots outliers, and the solid line in the box denotes the median of parameter estimates. **b** Internal uncertainty, which takes separate values for decisions from description (black) and experience (blue). **c** Social sensitivity, a weight that people put on social information irrespective of their uncertainty, for risky (red) and safe (dark blue) advice. **d** Reward sensitivity **e** temperature or randomness of choices.

and behavioural results align with the hypothesis that susceptibility to social influence in risky choice decreases across adolescent development, partly because people become less internally uncertain about the consequences of their choices as they develop. These results that internal uncertainty and social sensitivity together result in susceptibility to social influence add nuance to the understanding of the development of susceptibility to social influence and suggest that being internally uncertain about the utility of an outcome is a driver of social susceptibility that develops across adolescence.

## Discussion

We investigated the impact of uncertainty on the susceptibility to social influence in risky decision-making across adolescence. In a lottery experiment, we asked 166 participants aged 10–26 to decide between a safe or a risky option with or without social information favouring one of these options. We manipulated external uncertainty to be either low or high by asking for decisions from description and from experience, respectively. With a computational model that formalises social influence as Bayesian learning[25], we could differentiate our participants' internal uncertainty about the utility of a risky decision from external uncertainty that we manipulated in the experiment. With that model, we could also differentiate the effect of internal uncertainty on susceptibility to social influence from other, perhaps motivational, interindividual differences in people's social sensitivity. We hypothesised that susceptibility to social information for all participants would be greater when uncertainty was high. Further, we expected that adolescents would take more risks than people in other age groups, would be more susceptible to social information. We found little support for the preregistered hypothesis that adolescents would be more tolerant of uncertainty or use social information more than adults and younger children. We can confirm our preregistered hypotheses that uncertainty makes people more susceptible to social influence. Bayesian computational modelling suggests that decreasing internal uncertainty explains much of the age differences in susceptibility to social influence.

In line with many previous studies[50], the youngest participants decided for the risky option the most often, and this tendency was less strong the older our participants were. There was no evidence for a quadratic age component in our participants' propensity to take risks, which might imply

a special reward sensitivity of adolescents, or a peak in adolescents' risky decisions in our task. On average, all participants made fewer risky decisions when uncertainty in the experiment was higher, potentially demonstrating a case of "comparative ignorance", where people prefer less uncertain options when they can directly compare them to more uncertain options[51]. There was also no evidence of a quadratic age component in choosing risky under uncertainty. Bayes factors for quadratic age regressions, however, also provide no evidence for absence, as none is smaller than 0.33, which is often regarded as evidence in favour of the null hypotheses. Still, hypothesis H1 that adolescents would be tolerant to uncertainty and choose risk most often when uncertainty is high was not confirmed. Because of that, hypothesis H4, stating that because adolescents are uncertainty tolerant, their social susceptibility should not depend as much on external uncertainty, could also not be supported. A possible reason that in our study there was no evidence for an adolescent tolerance for uncertainty is that, unlike the current study, previous studies either involved no comparison to younger participants[33], or they required active exploration of outcomes[34,52–54]. Exploration-based learning relies on feedback processing, which is known to differ between age groups[55–57]. For example, adolescents tend to use feedback more when it confirms their previous beliefs[58]. In addition, Adolescents also weigh surprising negative feedback more than surprising positive feedback[56] and put less effort into searching for novel information[34]. All of these differences in response to feedback leave adolescents with another internal uncertainty than children or adults. This could result in overestimating the preference for making risky decisions in adolescents, while they might be simply more uncertain about what to do in an experiment that demands deciding between two options.

Adolescents' susceptibility to social influence is often thought of as a special social motivation or reward sensitivity in social contexts. When accounting for these factors in a computational model, we found that internal uncertainty was a separate factor (see Figs. S5 and S6 for a demonstration that their corresponding parameters are in principle orthogonal in the model) related to susceptibility to social influence in all participants that was less pronounced the older our participants were (Fig. 4). This age difference in susceptibility to social influence resonates with recent studies showing similar developmental trends in risky choice[43,49],

risk perception[59,60], prosocial behaviour[39,61], rule-following[62], belief formation[62] and trusting others[63]. Our model, which formalises susceptibility to social influence as Bayesian learning, offers a parsimonious explanation for some of the decreasing susceptibility to social influence that is shared across different domains. Internal uncertainty is likely an overarching factor that can account for susceptibility to social influence in such various domains and changes with age[62]. This means that to identify when young people are particularly susceptible to social influence, researchers or educators need to learn what it is that young people are uncertain about. If they are uncertain about how to get social approval, interventions could provide young people with predictable, less uncertain, means to do so[64], and by that reduce the temptation to engage in risky behaviour just to fit in.

Although uncertainty is an important mechanism of social influence, our data suggests that social influence and its development depend on the type of social information that participants see[65]. Our hypotheses that social influence would be generally stronger when (external) uncertainty is higher (H3) and that there is a quadratic age component in social influence (H2) can be confirmed, but only for safe social information. That is, younger participants were most influenced by safe social information, with an initially steep age-related decline of social influence which plateaued in adolescents. The impact of safe social information was particularly strong when external uncertainty was high; which is in line with previous work[35,36,43,49,66]. It could be that preference for safe choices, which is very prevalent in adults[67], may already exist in the youngest participants but only manifests in their choices with a small social nudge[48]. More complex social experiences of our participants could also contribute to this finding. For example, our youngest participants might be more inclined to follow safety-promoting social information because they are more frequently advised by others to avoid risks. As a result, younger children are more accustomed to listening to what others say and tend to associate non-compliance with more severe ramifications than our older participants. Not following safe social information is a stronger risk for younger participants in itself.

To our surprise, there was no main effect of safe social information, meaning that on average, participants were not more likely to choose safe when social information was in favour of the safe option. This could be driven by an additional preference spillover effect, where participants not only copied the specific choice but also adopted the other person's general preference for risk[68]. Because in our task, social information came from a risk-prone person, participants could have additionally shifted their preference as a whole[69], instead of only being sensitive to whether social information favoured risky or safe choices. This could explain why our oldest participants sometimes even opposed safe social information and chose risky instead.

## Limitations

This study has limitations. For instance, we cannot directly observe participants' thoughts about the advisor; these thoughts, however, may affect participants' susceptibility to social information. For example, even though we told participants that the advisor saw the same information as they did in decisions from experience, they still might believe that the advisor knows more or has different information. While the developmental decrease in participants' internal uncertainty remains unaffected by this, future research should assess participants' beliefs about the advisor's knowledge to further ascertain that susceptibility to social influence is about participants' own internal uncertainty about how to choose rather than participants' beliefs about others' expertise when people make decisions from experience.

A related limitation may be that we did not manipulate the advisor's identity or presence. This avoids confounding arbitrary characteristics of the advisor with a more general susceptibility to social influence, but on the other hand, research also suggests that young people attribute credibility to different sources of social information depending on the topic. For instance, adolescents' risk perception for recreational risks, such as skating or health risks, such as binge drinking, seems more easily influenced by peers than by adults[60]. Adults' advice, in turn, might be more impactful for financial risks[70].

Manipulating an advisor's identity in the task, or using vignettes with different topics, could help to shed light on which sources adolescents rely on most when they want to be surer about whether it is worth it to take a risk or not.

Finally, the conclusions here rely on one cross-sectional laboratory experiment. We cannot say what kinds of uncertainty exist in young people's real-world environments and how these uncertainties change with individual experiences that people make as they develop into adults. In real life risk-taking, internal uncertainty is highly individual, because it depends on individual experiences with risks and their probabilistic outcomes, which requires independent assessment[12]. Our work merely provides a starting point in asking for adolescents' internal uncertainty and, through that, better understand in which situations adolescents are at risk of being influenced by others.

In sum, we found that the development of susceptibility to social influence in risky choices depends on decreasing internal uncertainty. We showed that external uncertainty in the environment was related to increased internal uncertainty and social influence. Therefore, decreasing external uncertainty, for example, through establishing frequent feedback in educational settings or by providing predictable social rewards for adequate risks, will make the trade-offs of risk-taking clearer for young people. Knowledge of young people's internal uncertainty will give policymakers a tool to scaffold young people's risky behaviour adaptively[12]. Our work, for instance, suggests that communicating how others are likely to be as uncertain about prospects as they are, even though it may not always look like it, will help young people make more informed choices in social contexts. Knowledge about others' uncertainty helps to better reason about and act on social norms surrounding risky behaviours, which adolescents might sometimes misjudge[13].

By recognising the pivotal role of uncertainty underlying adolescent susceptibility to social influence, our findings underscore the need for a more nuanced perspective on the uncertainty that young people encounter on their way to adulthood. In efforts to understand adolescents' potentially maladaptive susceptibility to social influence, it is crucial to consider what teens feel uncertain about. This holds promise. Our findings demonstrate that internal uncertainty can make adolescents more susceptible to risky influences, but equally, it can open them up to positive social guidance.

## Data availability
All data is publicly available on: https://doi.org/10.5281/zenodo.16738297

## Code availability
JavaScript code with which the experiment was implemented, as well as all analysis code, can be found on: https://doi.org/10.5281/zenodo.16738297

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

## Acknowledgements

We thank Deborah Ain for editorial support, Chantal Wysocki for collecting the data, Antonio Amaddio for helping with SQL and PHP, Alexander Schakowski for discussions on the regressions and Jelka van Ross for illustrating the marble jar. Wouter van den Bos was supported by Open Research Area (ID 176), the Jacobs Foundation, the European Research Council (ERC-2018-StG-803338), and the Netherlands Organization for Scientific Research (NWO-VIDI016.Vidi.185.068). SC was supported by the Max Planck Society. This work was part of SC's dissertation work in the Max Planck UCL research school for computational psychiatry and ageing research (Comp2Psych). The funders had no role in study design, data collection and analysis, decision to publish or preparation of the manuscript.

## Author contributions

Simon Ciranka: conceptualisation, methodology, software, visualisation, writing—original draft, writing—editing. Wouter van den Bos: conceptualisation, writing—original draft, writing—editing, and supervision.

## Funding

## Competing interests

The authors declare no competing interests.
