## [Transparent Peer Review file · Communications Psychology]

Internal uncertainty impacts social information use in risky choice across adolescence

Corresponding Author: Dr Simon Ciranka

Version 0:

Decision Letter:

Dear Dr Ciranka,

Thank you for your patience during the peer-review process. Your manuscript titled "Internal uncertainty impacts social information use in risky choice across adolescence" has now been seen by 3 reviewers, and I include their comments at the end of this message. They find your work of interest but raised some important points. We are interested in the possibility of publishing your study in Communications Psychology, but would like to consider your responses to these concerns and assess a revised manuscript before we make a final decision on publication.

We therefore invite you to revise and resubmit your manuscript, along with a point-by-point response to the reviewers. Please highlight all changes in the manuscript text file.

Editorially, we consider it crucial that the methodological concerns regarding the experimental task, such as the choice of a riskier advisor and whether children can understand the task instructions, are thoroughly addressed in the revised manuscript. In addition, please make sure that the revised manuscript includes sufficient theoretical and methodological details and is well-structured to improve its readability.

Reviewer #1's suggestion on preregistration aligns with our journal policy. Please ensure all deviations from the preregistration are reported. All originally preregistered hypotheses and analyses should be included, unless scientifically unsound, in which case the deviation needs to be highlighted and explained. Please make sure that throughout the manuscript, it is clearly stated which analyses are preregistered and which are exploratory; clarity on this issue is particularly important in instances where the results of preregistered and exploratory analyses deviate. The full policy is here: <https://www.nature.com/commpsychol/editorial-policies/preregistration-policy>

I am attaching an Editorial Requests Table that details critical reporting requirements for the revised manuscript. Please attend to each item and ensure your manuscript is fully compliant. If your revised manuscript is not aligned with these requests on major issues, such as those concerning statistics, it may be returned to you for further revisions without re-review.

Please submit the following items:

- Revised manuscript
- Point-by-point response to the referees' comments
- Cover letter (as a separate document)

- <https://www.nature.com/documents/nr-reporting-summary.zip>>Nature Research Reporting Summary
- <https://www.nature.com/documents/nr-editorial-policy-checklist.pdf>>Editorial Policy Checklist
- Completed Editorial Request Table (attached).

via this link: Link Redacted .

Additional guidance is available in our style and formatting guide <https://www.nature.com/documents/commspsychol-style-formatting-guide-accept.pdf>>Communications Psychology formatting guide.

Best regards,

Troy Lui

Troy Lui, PhD
Associate Editor
Communications Psychology

REVIEWER EXPERTISE:

Reviewer #1: risk taking in adolescence, social influence
Reviewer #2: risk taking in adolescence, social influence
Reviewer #3: risk taking in adolescence, uncertainty, social influence

REVIEWER REPORTS:

Reviewer #1 (Remarks to the Author):

I appreciate the opportunity to review this important and interesting manuscript. Below, I provide specific comments regarding the theoretical framing, clarity of methods and analyses, and statistical reporting.

1. There were several points in the introduction where I disagreed with some of the content; specifically, I found it misleading, sensationalized, or not adequately the scientific content of the prior literature. Some examples...

*Freedom isn't intrinsically associated with risky behaviors. If that were true, we wouldn't see age effects in risky behavior

*Most teens don't "abuse" drugs, this seems like a pretty sensationalized claim. Teens definitely experiment with drugs and use casually, but I wouldn't call it abuse the way the authors characterize it here. Misleading without additional context.

*The phrase "giving them courage to learn" is not a scientific explanation. It is not tethered to established psychological theories of adolescent risk-taking--instead, it reads like a lay theory. The authors ought to clarify how this explanation aligns with existing frameworks on risk-taking and its developmental functions.

2. The discussion of different types of uncertainty (e.g., outcome variability, ambiguity of probabilities) is a welcome addition to the manuscript. However, the current terminology used to distinguish "external" and "internal" uncertainty is somewhat ambiguous and confusing. Renaming these constructs with more precise terms may enhance clarity.

3. It would be helpful if the authors explicitly confirm that all analyses have been disclosed and that any deviations from the pre-registration (if applicable) are clearly stated.

4. The task description is somewhat sparse and difficult to follow. Providing additional details, including a clearer explanation of the task structure and instructions given to participants, would improve comprehensibility.
5. What happens if the value of SI is changed from 1 to another value? I didn't see anything about this, though I could have very well missed it.
6. How do the authors ensure that 10-year-old children understood the task correctly? The following statements seem to imply the ten year olds may not have fully understood the task... "A positive interaction between expected value and age could mean that older people are more sensitive to the reward promised by the lotteries' expected value."
7. In Equation 3, the placement of the comma makes it appear as though there is a separate beta' parameter. Revising this would be help clarity.
8. The manuscript does not appear to report or justify the width of the credible intervals! This is a pretty notable omission, no??
10. It would be useful to include statistics assessing whether the sampler functioned appropriately. While these details do not necessarily need to be included in the main text, reporting on convergence diagnostics is needed to increase my confidence in the study.
11. The discussion of Bayes factors lacks inferential criteria. Bayes factors allow for graded inference, but it is unclear how the authors are applying them in this context. Clarifying the interpretative framework for Bayes factor results would strengthen the manuscript's conclusions.

Reviewer #2 (Remarks to the Author):

In a sample of 166 participants aged 10-26 years, this preregistered study used experimental tasks and computational modeling to examine age differences in risky decision-making and social influence as a function of the external uncertainty of the experimental environment (i.e., decisions from description vs. experience), internal uncertainty about decision-making, and social sensitivity. Results revealed age-related declines in internal uncertainty and susceptibility to social influence across adolescence. Susceptibility to social influence was greater when external uncertainty was higher, especially for younger participants. The results are very interesting and will advance the field of adolescent social influence by highlighting internal and external uncertainty as alternative mechanisms, moving beyond the traditional emphasis on reward or social sensitivity. Below, I offer a few suggestions for improving the manuscript:

Major:

Could the authors please define social sensitivity in the Introduction?

p. 3. "This uncertainty is part of someone's assumptions about the world, prior to seeing social information. If one's prior assumptions about the utility of their decisions are uncertain, they will be more inclined to listen to what others say and update their beliefs about the utility of the decision after seeing social information." Should this be the utility of the social information for guiding their decisions, not the utility of the decision itself? It would also be helpful for utility to be defined earlier in this paragraph.

The Introduction should include more background on age differences in uncertainty to justify hypotheses. Are hypothesized age differences specific to external and/or internal uncertainty? Can the authors preview the competing hypotheses tested by the alternative cognitive models (e.g., model parameters added)?

p. 6. "Participants knew nothing about the advisors' identity, only that we tried to find an advisor matched to their risk preference, which we did by finding an advisor who, on average, made 15 more risky decisions (20%) than the participants in the first half of the experiment."

- What were participants told about the advisor's role and choices in the experiment? Are there post-task manipulation checks to test whether the participants believed the advisor was real or a credible source of information during decision-making? Did participants believe that they were observing the same advisor during the decision from description and decision from experience conditions?

- I commend the authors' effort to match the advisor's choices to participant's initial choices. However, an advisor making 20% more risky choices than the participant seems less like a close match and more like a notably higher level of risky decision-making to observe. Can the authors clarify why the 20% more risky choices threshold was chosen for the "matched" advisor, and why a similar match wasn't made for safer choices as well?

- For the Social condition, I do not think seeing how someone else chooses between two jars can be called "advice," as the latter implies intentional guidance or communication meant to influence another's decision. I suggest rewording "advice" to something like "observed choices."

The manuscript currently describes social information as coded '-1' for none, '0' for safe, and '1' for risky advice (p. 8). This coding could be a bit confusing, as it appears to make 'safe advice' the reference category rather than 'no advice,' which is more commonly used as a neutral baseline. It might help readers if this were clarified, or if the coding were adjusted (e.g., '-1' for safe, '0' for none, and '1' for risky) to make the comparisons and interpretation of effects more intuitive. (You can then

use contrast coding to directly compare risky vs. safe advice).

Figure 4 caption: Is susceptibility to social influence (difference in prior p and posterior p') the same as social impact? Why use the absolute difference score $|p-p'|$ instead of the valenced difference score? For Figure 4b. Is internal uncertainty ($1/k$) computed separately for decisions from description vs. experience (is that what the blue and gray colors indicate)?

To provide a better overview of findings across your sample, it would be helpful to provide in the main manuscript the overview of the (model-free) regression results (i.e., move Table S1 to main manuscript), as well as descriptives (means, SD, ranges) of key variables of interest from both the behavioral model (e.g., like those shown in Figure S2) and computational model (Figure 4).

To give readers a clearer overview of the findings across your sample, I recommend including the model-free regression results (currently in Table S1) in the main manuscript. It would also be helpful to present descriptive statistics (means, SDs, ranges) for key variables from both the behavioral model (e.g., as in Figure S2) and the computational model (e.g., Figure 4) within the main text.

p. 19. "In sum, susceptibility to social influence was more pronounced in younger participants than in adults for two reasons." At the end of the Results section, higher internal uncertainty and higher social sensitivity in younger participants was suggested as the two reasons for the observed age-related differences in susceptibility to social influence. Has this mediation pathway (age > internal uncertainty/social sensitivity > susceptibility to social influence) been formally tested? If the mediation model isn't tested or isn't significant, I suggest using less causal language in the Results and Discussion sections.

I appreciated the interesting, well-written discussion and broader impacts of the study results, especially on potential explanations underlying the observed null quadratic age findings.

Minor:

Typo: p. 4. "Finally, adolescents internal uncertainty could be higher, so even if they would do not fear strong negative consequences of a risk they take, they still could be more uncertain than adults if they really want to be part of that group (Reiter et al., 2021)..."

p. 13. "(b_EV=4.12, CI = [4.00, 4.24], BF > 100; figure S2a)..." Can the authors please clarify what BF values are?

I suggest rewording this result to better highlight the significant age effect in social sensitivity, since it currently reads like there were no age differences: "In line with previous research, participants were less sensitive to safe (b_age = -0.15, CI = [-0.19, -0.12], BF > 100; b_age2 = -0.12, CI = [-0.16, -0.08], BF > 100 figure 4c; *ψsafe*) and risky (b_age = -0.12, CI = [-0.16, -0.07], BF > 100; b_age2 = -0.12, CI = [-0.16, -0.08], BF > 100 figure 4c; *ψrisky*) advice across adolescence" (p. 17).

Reviewer #3 (Remarks to the Author):

I would like to thank the editor and the authors for giving me the opportunity to review this paper, which I found highly interesting. The question raised by this paper - the effect of internal/external uncertainty on susceptibility to social influence during adolescence - is of major importance, as it addresses a significant issue about social influence during adolescence, with far-reaching implications. The authors' methodology and analyses are perfectly suited to tackle this critical question, demonstrating a rigorous approach and real expertise in the field. My main reservation about the paper lies in its didactic qualities: the authors should make more efforts to enhance its clarity and accessibility, particularly regarding the hypotheses, the analyses conducted and results. Moreover, I do not entirely agree with the interpretation of one of the key findings of the study regarding the decrease with age in the tendency to follow peer advice. As I will elaborate later, the authors do not consider a central element in this result, whose implications, in my opinion, are nonetheless significant and even support the authors' conclusion regarding the decrease in internal uncertainty with age. Even though, this does not undermine the major contribution of this study.

Below, I offer the authors suggestions for further improving their manuscript.

INTRODUCTION

General comments: The authors present a relevant and comprehensive review of the existing literature. I particularly appreciated the emphasis placed on the adaptive value of risk-taking and peer influence during adolescence. These aspects, while relatively recent in research discussions, are crucial for understanding developmental processes and decision-making in this age group.

Comment 1:

The authors put forward three explanatory hypotheses to account for the developmental difference in adolescents' social susceptibility. The first suggests greater susceptibility and stronger conformity towards peers. The third posits higher internal uncertainty, making adolescents more easily influenced. As for the second, it requires clarification: I understand that, in a social context, external uncertainty leads to greater internal uncertainty, but in that case, I struggle to see how it differs from the third explanatory hypotheses?

In the discussion (l. 379-381), the authors conclude: "That is, previous results are compatible with our modelling 378 result that internal uncertainty, that declines with age, is a mechanism behind susceptibility to social influence. 379 This suggests

that focusing only on developmental changes in reward sensitivity or the need to belong is 380 insufficient to understand adolescent susceptibility to social influence." I wonder if it would not be more coherent to distinguish in the introduction the following three explanatory hypotheses: 1) greater sensitivity to rewards, 2) the importance of conformity and the sense of belonging, 3) higher internal uncertainty.

Comment 2:

Some concepts need further clarification. For instance, while the authors clearly differentiate the terms "social sensitivity" and "susceptibility to social influence" in the introduction, a conceptual definition of these two terms seems essential to help readers fully grasp the distinction. Providing clear explanations would strengthen the paper's accessibility and ensure a better understanding of how these concepts are applied within the study.

HYPOTHESES

General comments:

The hypotheses were preregistered, which is highly commendable. However, I find it unfortunate that the hypotheses are much more clearly detailed on OSF than in the article itself. In striving for conciseness, the authors do not provide readers with enough detail to fully understand the hypotheses.

Comment 1: Regarding the first hypothesis (more risky choices among adolescents in decision-from-experience rather than decision-from-description due to greater tolerance for uncertainty), I would like to share a reflection concerning the results of previous studies (e.g., work by Tymula, Blankenstein, etc.). I wonder to what extent the increase in risky choices in conditions comparing a safe option to an ambiguous gamble might be driven by greater sensitivity to rewards rather than a greater tolerance for uncertainty per se. Indeed, the ambiguous gamble offers a much more attractive reward than the safe option. I am fully aware that the same applies to the non-ambiguous gamble and that reward sensitivity should also lead adolescents to prefer the non-ambiguous gamble. However, I see a fundamental difference here: since probabilities are not available (or only partially) in the ambiguous condition, it is likely that the remaining information—namely, the amount associated with each option—becomes more salient, leading adolescents to choose the ambiguous gamble more frequently given greater reward sensibility. This might explain why adolescents appear as ambiguity averse as adults in tasks inspired by Ellsberg's urns, which directly compare risky and ambiguous options with equivalent associated rewards (see Osmont et al., 2022). I would be curious to hear your thoughts on this issue.

Comment 2: Regarding the second hypothesis (greater sensitivity to peer opinions in decision from experience compared to decision from description), the authors should support their hypothesis by referencing the article by Osmont et al. (2021). This study shows that adolescents refer to the prior choices of their peers in a risk-taking task (BART) only in the uninformed condition (corresponding to decision from experience) and not in the informed condition. Furthermore, this paper demonstrates a greater influence of produced peer choices compared to risky peer influence. The authors could therefore refer to it in several parts of the introduction (e.g., paragraph line 49).

Osmont, A., Camarda, A., Habib M., & Cassotti, M. (2021). Peers' choices influence adolescent risk-taking especially when explicit risk information is lacking. *Journal of Research on Adolescence*, 31 (2), 402-416. <https://doi.org/10.1111/jora.12611>

Comment 3: Hypothesis 4 needs to be clarified. Referring to the preregistration on OSF, I understand that adolescents, being more tolerant of uncertainty, would exhibit less effect of uncertainty level on their sensitivity to peer advice: they would be sensitive to peer advice in both experience-based and description-based decisions, while adults would be more sensitive in experience-based scenarios. However, following this reasoning, I wonder: if external uncertainty leads to increased internal uncertainty, and internal uncertainty in turn enhances susceptibility to peer advice, then wouldn't adolescents, being more tolerant of uncertainty, experience less internal uncertainty? Consequently, wouldn't this make them less susceptible to peer advice compared to adults? This appears somewhat contradictory with previous works, and I would be curious to hear your perspective on how this interplay is reconciled within the framework of the hypothesis.

METHOD

General comments: The methodology is clearly presented and aligns well with the objectives. I only have a few clarifications to highlight and some of the choices made in the design might need additional justification for the sake of transparency.

Comment 1: There are fewer girls than boys in the study. Is this distribution consistent across different age groups? If not, it should be addressed as a limitation. Figure S1 perfectly reports the distribution of age and stages of puberty. I suggest a similar type of figure illustrating the gender distribution by age in the supplementary materials.

Comment 2: In the decision from experience condition, participants learn by observing 9 samples. Why exactly 9 samples were chosen? It would be interesting to know if this number was determined based on pilot studies or theoretical considerations. As for the outcomes of the samples, clarification is needed. Are the results pre-determined (fixed) to ensure consistency across participants, or are they randomly generated? Providing this information would enhance the transparency and reproducibility of the study.

Comment 3: Regarding the advisors' choices, the procedure lacks clarity. The justification for presenting the advisor as

having a similar risk-taking level to the participant is only mentioned in the discussion. Are the advisor's advices based on the actual responses of a participant who previously completed the task? And why the advisor made 15 more risky decision in the first half of the experience ? It would be helpful to specify exactly how this concept of "matched with their risk preference" is explained to the participants. I wonder how such an abstract notion is perceived and understood by an adolescent of 10 years.

RESULT

General comments: My main concerns refer to the didactic dimension of the presentation of results. Even though I am not an expert in the statistical models used in this study, understanding the results required considerable effort on my part. I found myself having to go back and forth between the article's text and the supplementary material figures, which I occasionally found inconsistent. This complexity applied even to simpler interaction analyses. Greater cohesion between the text, figures, and supplementary material would certainly improve accessibility and readability.

Comment 1: l.235: The authors highlight an unexpected effect of the advisor's cautious advice: more risky choices. Could this be explained by conformity toward the advisor, who could be generally perceived as more risk-seeking, even during their cautious advice? Participants might conform to their general perception of the advisor's risk preference rather than to each piece of advice individually. The reference to the 15 riskier choices made in the first part brings me to this idea, although I may have misunderstood this point in the methodology.

Comment 2:

l. 250: The term "experience condition" is indeed potentially confusing, as it more naturally refers to the decision-from-experience condition.

l. 253: The authors mention an interaction between the linear age term and the level of uncertainty (experience versus description). They state "participants' propensity to choose risky decreased with age when making decisions from experience ". However, what about the decision from description condition? In Figure S2i, the slope appears less pronounced compared to the decision from experience condition, but there is still a noticeable decrease with age.

Comment 3:

l. 255-263 (and after): I am not entirely convinced by the way the authors operationalize susceptibility to peer advice in their results. The authors interpret susceptibility solely based on the safe advice trials (or risky advice). However, I believe that the discussion about susceptibility to the advisor's advice relies on a comparison between the advice condition (sage or risky) and the control condition (none).

For instance, based on Figure S2h, the following questions arise:

- Does the safe/experience condition differ from the None/experience condition?
- Does the risky/experience condition differ from the None/experience condition?
- Does the safe/description condition differ from the None/description condition?
- Does the risky/description condition differ from the None/description condition?

Comment 4:

l. 264-274: Concerning H3, the results are somewhat difficult to follow.

Regarding the decision from description condition, it is evident that there is sensitivity to risky advice, which remains consistent across all ages (as observed in the comparison between the "risky" and "none" lines).

However, for the decision from experience condition, I don't entirely align with the authors' conclusions. Based on Figures S2k and, more notably, S2i, it appears that younger participants tend to follow cautious advice, resulting in less risk-taking compared to the "none" condition. On the other hand, for older participants, I would have concluded to an unexpected effect of cautious advice—it seems clear that there is an increase in risky choices compared to the control condition.

Comment 5:

l. 282-285: Concerning H4, the results are also difficult to follow : "with increasing age, advice favouring the safe jar was related to more risky decisions when participants made decisions from experience, where external uncertainty was higher ($b_{\text{safe} \times \text{experience} \times \text{age}} = 283.045$ CI = [0.1, 0.8], BF = 52.00; figure S2m), suggesting that younger participants were steered away from choosing risky more strongly than older participants when external uncertainty was high."

If, with age, risky advice led to more risky choices, the blue curve would be ascending. However, it remains slightly descending. Are the authors rather referring to the quadratic effect of age (f.S2n instead of f.S2m)?

In figure f.S2m we observe that older participants in the decision from experience condition oppose their peers' advice and make more risky choices compared to the control condition.

Comment 6:

l. 286-289: "In sum, behavioural results indicate that older participants choose more cautiously than younger ones when 286 there is high external uncertainty requiring decisions from experience, with no adolescent peak in risky choices. 287 Overall, participants were more susceptible to social influence when advice favoured the safe, but not the risky 288 jar, when external uncertainty was high and required decisions from experience." The authors provide a summary of the main results in connection with their hypotheses, which is indeed necessary. However, I only partially agree with their conclusions. While I align with the authors' first statement, I disagree with the second. It seems to me that exposure to risky advice influences participants' choices in the decision-from-experience situation, regardless of their age, by increasing the number of risky choices. On the other hand, when it comes to cautious advice, younger participants tend to follow this advice, whereas older

participants seem to oppose it.

DISCUSSION

General comments: The discussion presented here is well-conducted and effectively highlights what I consider to be the most important result of this study: the decrease in internal uncertainty during adolescence, which appears to be a central mechanism for sensitivity to social influence. However, several points must be further discussed.

Comment 1: I would have liked the authors to further discuss their first hypothesis. While they conclude from their results a decrease in risky choices with age in the decision from experience condition, they do not address the decision from description condition. Even though the slope is steeper for the decision from experience condition (figure S2i and j), it appears a quite similar decrease for the decision from description condition. Therefore, these results do not seem to truly validate H1 or support previous studies. Personally, I am quite convinced that adolescents experience greater uncertainty, but not that they are more tolerant of this uncertainty (as noted in Hypothesis comment 1). Although I agree with Tymula's idea that tolerance to uncertainty holds essential adaptive value for adolescents, I believe that most of them are already highly intolerant of it.

Comment 2: Once again, the result indicating that older participants decide against the advisor cautious choices in the decision from experience condition seems to be ignored by the authors. In my view, this finding further reinforces the aforementioned conclusion. Younger adolescents are more likely to follow their peers' cautious choices due to greater internal uncertainty, unlike young adults. Thus, internal certainty, which increases with age, appears to be crucial for resisting peer influence and even opposing their advice. This point warrants further discussion.

Comment 3: l. 409-411: "Since we matched advisors to our participants' choices, the tendency to use risky advice more, which 409 we observe consistently in participants older than 13 but not in younger participants, may reflect an emerging 410 ability to discern advice that feels useful to participants from advice that does not (Moses-Payne et al., 2021)." Another point should be further discussed. The results of this study also challenge the stereotypical view of adolescents as particularly vulnerable to risky peer influence and therefore more likely to take risks to follow their peers. Here, we note that susceptibility to the advisor's risky influence is not age dependent. Only sensitivity to cautious influence is higher in early adolescence. Thus, adolescents do not follow the advisor's risky advice more often; instead, they are more sensitive to their cautious advice. This point is also supported by the findings of Osmon et al. (2021), suggesting that peers' prior cautious choices have more impact on adolescents' risk-taking decisions than previously risky choices in ambiguous situations. Adolescents follow their peers' risky choices only when these align with the trial's risk level, also indicating discernment in facing peer influence.

Comment 4: l. 397-430:

In this paragraph, the authors should reference the comparative ignorance hypothesis:

Fox, C. R., & Tversky, A. (1995). Ambiguity Aversion and Comparative Ignorance. *The Quarterly Journal of Economics*, 110(3), 585-603. <https://doi.org/10.2307/2946693>

TYPOGRAPHICAL REVISIONS

l. 84-85: The sentence seems incomplete.

l. 32 : "und" instead of "and"

Communications Psychology is committed to improving transparency in authorship. As part of our efforts in this direction, we are now requesting that all authors identified as 'corresponding author' create and link their Open Researcher and Contributor Identifier (ORCID) with their account on the Manuscript Tracking System prior to acceptance. ORCID helps the scientific community achieve unambiguous attribution of all scholarly contributions. You can create and link your ORCID from the home page of the Manuscript Tracking System by clicking on 'Modify my Springer Nature account' and following the instructions in the link below. Please also inform all co-authors that they can add their ORCIDs to their accounts and that

they must do so prior to acceptance.

Version 1:

Decision Letter:

Dear Dr Ciranka,

Your manuscript titled "Internal uncertainty impacts social information use in risky choice across adolescence" has now been seen by our reviewers, whose comments appear below. In light of their advice I am delighted to say that we are happy, in principle, to publish a suitably revised version in *Communications Psychology*.

We therefore invite you to revise your paper one last time to address the remaining concerns of our reviewers and a list of editorial requests. At the same time we ask that you edit your manuscript to comply with our format requirements and to maximise the accessibility and therefore the impact of your work.

EDITORIAL REQUESTS:

SUBMISSION INFORMATION:

OPEN ACCESS:

*** TRANSPARENT PEER REVIEW:** *Communications Psychology* uses a transparent peer review system. On author request, confidential information and data can be removed from the published reviewer reports and rebuttal letters prior to publication. If you are concerned about the release of confidential data, please let us know specifically what information you would like to have removed. Please note that we cannot incorporate redactions for any other reasons.

*** CODE AVAILABILITY:** All *Communications Psychology* manuscripts must include a section titled "Code Availability" at the end of the methods section. We require that the custom analysis code supporting your conclusions is made available in a publicly accessible repository at this stage; please choose a repository that generates a digital object identifier (DOI) for the code; the link to the repository and the DOI must be included in the Code Availability statement. Publication as Supplementary Information will not suffice.

* DATA AVAILABILITY:

All *Communications Psychology* manuscripts must include a section titled "Data Availability" at the end of the Methods section. More information on this policy, is available in the Editorial Requests Table and at <http://www.nature.com/authors/policies/data/data-availability-statements-data-citations.pdf>.

Link Redacted

Best regards,

Troby Lui

Troby Lui, PhD
Associate Editor
Communications Psychology

REVIEWERS' COMMENTS:

Reviewer #1 (Remarks to the Author):

The authors have addressed all my concerns.

Reviewer #2 (Remarks to the Author):

I appreciate the authors' extensive revisions and responses to reviewer comments and believe the current paper will make an exciting contribution to the adolescent social influence literature. The comprehensive revisions, especially in clarifying the definitions of key terms and hypotheses and variables' coding and interpretation, address many of the previously raised concerns. Two minor suggestions remain to improve the clarity of the Introduction:

Knight 1921's definitions of epistemic and aleatoric uncertainty were a valuable addition to the theoretical foundation and chosen taxonomy presented in the Introduction. I think the authors' reviewer response noting that some types of uncertainty are reduceable through learning than others, whereas others are not is conceptually important to include in the main manuscript. To that end, I suggest a minor edit to the following sentence (p. 3, Introduction): "Thus, there is internal uncertainty in someone's beliefs about the utility of a choice that can be reduced through learning and experience and there is external uncertainty that is a feature of the unpredictability of the environment itself that cannot be reduced (Knight, 1921)."

In general, the revised hypotheses are easier to understand, but I think the reference groups for H3 and H4 remain unclear and should be made more explicit. For H3, adolescents are most influence by social information compared to children and adults? Compared to no social information? For H4, adolescents' susceptibility to risky social influence should be less affected by our experimental manipulations of uncertainty compared to children and adults?

Reviewer #3 (Remarks to the Author):

I would like to sincerely thank the authors for the quality and precision of their responses to each of the comments raised by the three reviewers. Their thorough work has helped clarify key points in the article, made the results and conclusions more understandable, and highlighted the valuable contributions of this study. I am therefore in favor of the publication of this article. My only remaining concern is that Figure S2, which is referenced repeatedly in the results section, has been placed in the supplementary materials. It serves as an essential tool for understanding the findings and, in my view, should be included in the main article (at least in part) to avoid requiring readers to constantly navigate between the paper and the supplementary material. Once again, thank you for inviting me to participate in this review, which has been stimulating and enriching.

Reviewer #1 (Remarks to the Author):

I appreciate the opportunity to review this important and interesting manuscript. Below, I provide specific comments regarding the theoretical framing, clarity of methods and analyses, and statistical reporting.

Thank you for the positive evaluation and for making the effort to share your insights on how to improve the paper.

1. There were several points in the introduction where I disagreed with some of the content; specifically, I found it misleading, sensationalized, or not adequately the scientific content of the prior literature. Some examples...

We agree that at some points we could have tempered our formulations. We appreciate the opportunity to write a more solid manuscript.

*Freedom isn't intrinsically associated with risky behaviors. If that were true, we wouldn't see age effects in risky behavior

We agree, freedom is not intrinsically associated with risky behaviour. We already tried to weave that into the submitted version, where we wrote "this freedom *can* be risky". But this was not enough. We elaborate on it now, where we add afterwards in the introduction:

New opportunities do not necessarily lead to more risky behaviour, but novelty can encourage risk-taking. For instance, risk-taking among US citizens appears to peak only after adolescence, when young adults are in their early twenties and have more opportunities (Willoughby et al., 2021). It is, however, during adolescence that many people engage in risky behaviours for the first time, such as skipping school (Duell et al., 2018), driving recklessly (Yellman et al., 2020) [...]

*Most teens don't "abuse" drugs, this seems like a pretty sensationalized claim. Teens definitely experiment with drugs and use casually, but I wouldn't call it abuse the way the authors characterize it here. Misleading without additional context.

We again agree, and now write "experiment with."

*The phrase "giving them courage to learn" is not a scientific explanation. It is not tethered to established psychological theories of adolescent risk-taking--instead, it reads like a lay theory. The authors ought to clarify how this explanation aligns with existing frameworks on risk-taking and its developmental functions.

Thank you for raising this point. Again, we agree. We now write "it motivates them to [...]" The formulation "courage" is indeed inappropriate given the evidence, because it suggests they lack courage before, which would be a strong claim and not what we meant to convey.

2. The discussion of different types of uncertainty (e.g., outcome variability, ambiguity of probabilities) is a welcome addition to the manuscript. However, the current terminology used to distinguish "external" and "internal" uncertainty is somewhat ambiguous and confusing. Renaming these constructs with more precise terms may enhance clarity.

Here as well, we agree that there is a problem. We struggled with finding the right way to introduce uncertainty. Finding the right terms was not trivial in this manuscript and in our attempt to be clear, we tried different taxonomies, but none of them were free from problems, completely expressed what we refer to in this work, or is agreed on across fields. After some debate, we concluded that another terminology would not be the solution, and we made our best effort to write more clearly instead. In the following, we briefly elaborate on the taxonomy issue.

The issue we faced in writing this manuscript is that there appears to be no universally accepted taxonomy of uncertainty. This taxonomy issue is even exacerbated for us because we actually do make an explicit distinction between external and internal uncertainty, meaning there are unknowns in the external world which can lead someone to be unsure (internally uncertain) about which decision to make, which is not part of the most widely spread taxonomies in the adolescent literature.

We are aware of various taxonomies of uncertainty. Some only use one “uncertainty” and refer to any unknown as uncertainty (Juechems et al., 2021; Tversky & Kahneman, 1992). Others distinguish between two types of uncertainty: situations of uncertainty (confusingly), where outcome probabilities are unknown, and situations of risk (which we felt was additionally confusing when also talking about risk-taking hence we added a footnote), where outcome probabilities are known (Kozyreva et al., 2019; Volz & Gigerenzer, 2012). Another taxonomy distinguishes three types of uncertainty: risk, uncertainty and ambiguity, where ambiguity is information that is “known to be missing”. Sometimes this ambiguity is also referred to as “radical” uncertainty (which again has different subtypes) (Kay, 2020). This broader taxonomy introduces a novel category, ambiguity, to the previous distinctions. Still, it describes an environmental, or external, uncertainty, rather than a state of uncertainty in the decision maker (Levy et al., 2010).

Another taxonomy distinguishes five (sensory, state, rule, and outcome uncertainty as well as additional “noise”) (Bach & Dolan, 2012) types of uncertainty, where individual and uncertainty in the environments are intermixed but without making explicit which kind of uncertainty refers to environments and which are internal to the mind.

One type of uncertainty that clearly seems not to be part of the environment is people’s uncertainty about their preferences, which has previously been referred to as “taste” (Moutoussis et al., 2016),(Moutoussis et al., 2016) or “preference” (Reiter et al., 2021),(Reiter et al., 2021) uncertainty. We explicitly mention preference uncertainty in the discussion, as this seems to be a type of uncertainty that is definitely internal to the individual, but we did not model preference uncertainty as it was done before (with a probability distribution over a time-preference parameter in a computational model) so we decided against this formulation as well.

The taxonomy that we found most useful in our work differentiates between “epistemic” and “aleatoric” uncertainty (Knight, 1921). Epistemic uncertainty is inherent in the individual and reducible through learning; aleatoric uncertainty refers to unknowns in the environment that cannot be reduced. “Knightian” uncertainty is sometimes used to express the superposition of the two. We would be happy to consider this taxonomy consistently in another revision if this appears better, but we still had the impression that “internal” and “external” may be easier to follow for a more diverse audience.

We are grateful for this comment and took it as an opportunity to edit the introduction and discussion, where we make this connection to the work of Knight more explicit and qualify, also in response to comments from the other reviewers, we have improved our hypotheses about uncertainty. Still, we hope for your understanding that a complete discussion of uncertainty terminology or attempting to find the correct terminology of the types of uncertainty is beyond the scope of this article, which is mostly concerned with the mechanism of social influence in adolescent risky choice. In response to the reviews here, we edited and actually deleted much of our elaboration on uncertainty, as it ultimately made our arguments harder to follow rather than easier.

In the revised paper we, make clear that our distinction aligns with the time-honoured work of Frank Knight:

The uncertainty in someone's beliefs, is not only shaped by experience, but also by the statistical structure of the environment. Sometimes aspects of the environment can be learned, and through repeated exposure and feedback, individuals can gradually reduce this uncertainty, refine their beliefs and form more stable beliefs about potential outcomes of their decisions (Bach & Dolan, 2012). Often times, however the environment is hard to predict, making it difficult to form confident expectations and reduce uncertainty about the utility of a decision, regardless of experience. In other words, people and environments can differ in their state of uncertainty. Thus, there is internal uncertainty in someone's beliefs about the utility of a choice and there is external uncertainty that is a feature of the unpredictability of the environment itself (Knight, 1921).

In summary, after some debate, we concluded that a distinction between internal and external uncertainty is the most useful for our paper; however, we needed to situate it more effectively within our paper. We hope that our arguments are a convincing case in favour of our terminology and appreciate the encouragement to think more clearly about uncertainty.

3. It would be helpful if the authors explicitly confirm that all analyses have been disclosed and that any deviations from the pre-registration (if applicable) are clearly stated.

Agreed. We are committed to transparency. Thus, we have included and reported all conducted analyses in this manuscript. Additionally, we acknowledge in the main text that our preregistered hypotheses were vague, and we did not specify in the preregistration which regression results would relate to which hypothesis.

In the methods section, we now also spell out one deviation from our analysis plan, which is that we preregistered we would code social information as 1 (when there is social information) and 0 (when there is no social information). However, this would have been an unfortunate analytic choice, as it would have made us blind to social influence for risky and safe social information. We added a section that reads:

Deviation from the preregistered regression model. We preregistered to code social information as 1 (when there is social information) and 0 (when there is no social information). However, this was not an ideal specification, as it would have made us unable to detect differences between the influence of risky and safe social information and these opposing social forces might even have cancelled each other out. We also note that while we preregistered a statistical model (see methods), we failed to preregister inference criteria and how the results of the model would lead us to accept or reject these hypotheses. We interpret the regression results according to general conventions in the literature and added marginal effects plots to aid their interpretation.

For the computational modelling, there was no analysis plan; however, the modelling was stated as "exploratory" in the preregistration, partly due to a lack of protocol regarding computational modelling and preregistrations. We now write in the revised manuscript:

To understand the relationship between age and the parameters of the cognitive model, we estimate a multivariate Bayesian linear regression and predict parameters of the cognitive model with linear and quadratic polynomials of our participants' age. We preregistered the computational model as exploratory analyses and did not specify particular hypotheses here.

4. The task description is somewhat sparse and difficult to follow. Providing additional details, including a clearer explanation of the task structure and instructions given to participants, would improve comprehensibility.

Thank you for pointing this out. We have addressed this by adding additional details to our task description and restructuring the methods section to introduce conceptually related aspects together. We also added subheaders for different aspects of the task, which additionally helps to provide more structure. Additionally, in response to a comment 6 raised in this review and another point by reviewer 2, we have included the instructions as seen by the participants in the supplement to enhance comprehensibility.

5. What happens if the value of SI is changed from 1 to another value? I didn't see anything about this, though I could have very well missed it.

With a higher value of SI, people would be more easily persuaded to choose what the social information showed. We have now made it clear in the methods section that SI is a dummy variable indicating whether social information favoured the risky or the safe option.

The parameter that allows for differential influence of that social information, effectively changing SI, is ψ . The effect of ψ is depicted in the panels of figure 2. We now write:

In Eq 4, SI is a dummy variable that codes whether the other chose risky or safe, but participants may still differ in their social sensitivity, irrespective of their internal uncertainty [...]

6. How do the authors ensure that 10-year-old children understood the task correctly? The following statements seem to imply the ten year olds may not have fully understood the task... "A positive interaction between expected value and age could mean that older people are more sensitive to the reward promised by the lotteries' expected value."

Thanks for this careful reading. We had an experimenter reading the instructions to the kids, and the experimenters were instructed to ask if the kids understood what they were supposed to do. For clarity, and also in response to reviewer 2, we added the following statement to the methods section.

[...] To ensure that even the youngest participants understood the task, an experimenter read out detailed instructions, asked for a comprehension check, and moderated a short practice round before participants started the task. Further there were three built in comprehension check in the instructions, that participants needed to pass before being able to proceed with the experiment (see supplementary material for screenshots of the instructions).[...]

We want to note here however, that younger people quite commonly make fewer EV-maximising choices in such tasks (Blankenstein et al., 2016; Braams et al., 2019; Giron et al., 2023; Meder et al., 2021; Palminteri et al., 2016; Schulz et al., 2019), often framed as increased exploration or a higher temperature parameter, which is consistent with our finding. Despite this, the youngest kids in this study also did not make their decisions randomly. Thank you for the chance to improve the credibility of this manuscript for the reader.

7. In Equation 3, the placement of the comma makes it appear as though there is a separate β' parameter. Revising this would be help clarity.

Thanks for pointing to this potential ambiguity! We now write $p = \alpha / (\alpha + \beta)$, and it no longer appears as if there is a separate β' parameter.

8. The manuscript does not appear to report or justify the width of the credible intervals! This is a pretty notable omission, no??

It is indeed a gross and surprising omission; thank you for pointing it out to us! We now clarify that we report the 95% credible interval and add 95% to every reporting of the credible interval.

10. It would be useful to include statistics assessing whether the sampler functioned appropriately. While these details do not necessarily need to be included in the main text, reporting on convergence diagnostics is needed to increase my confidence in the study.

This was another omission by us, and we now also write in the main text:

Regressions were run with 10,000 iterations of four chains. The convergence of the Markov chains was assessed by consulting the Gelman-Rubin statistic, which was 1 for all reported regressions indicating convergence.

11. The discussion of Bayes factors lacks inferential criteria. Bayes factors allow for graded inference, but it is unclear how the authors are applying them in this context. Clarifying the interpretative framework for Bayes factor results would strengthen the manuscript's conclusions.

Thank you, this is yet another omission! We now write in the manuscript:

We report Bayes factors (BF) as the Savage Dickey density ratio between priors (Cauchy priors and posterior estimates for the regression weights. In general, Bayes factors that are larger than 3 are considered substantial evidence in favour of the alternative hypothesis. Bayesfactors smaller than 0.3 are considered substantial evidence in favour of the prior (the null hypothesis). Bayesfactors between 0.3 and 3 are considered inconclusive.

We also took this opportunity to adhere to better practices in reporting Bayes factors. Additionally, while revising, it became apparent to us that the way we calculate Bayes factors (using the hypotheses function implemented in brms, the R package we used for regressions) is not a proper Bayes factor and should be better named an evidence ratio. This method required us to report separate Bayes factors for the null and alternative hypotheses, which is a shortcoming that another method can avoid.

We now use another R package (bayestestR) to calculate proper Bayes factors; therefore, the Bayes factors in our paper have changed in value, but not in the conclusions they afford, in this revised manuscript. So, thank you very much for the chance to adjust our manuscript to use a more cutting-edge standard in calculating and reporting Bayes factors.

Reviewer #2 (Remarks to the Author):

In a sample of 166 participants aged 10-26 years, this preregistered study used experimental tasks and computational modeling to examine age differences in risky decision-making and social influence as a function of the external uncertainty of the experimental environment (i.e., decisions from description vs. experience), internal uncertainty about decision-making, and social sensitivity. Results revealed age-related declines in internal uncertainty and susceptibility to social influence across adolescence. Susceptibility to social influence was greater when external uncertainty was higher, especially for younger participants. The results are very interesting and will advance the field of adolescent social influence by highlighting internal and external uncertainty as alternative mechanisms, moving beyond the traditional emphasis on reward or social sensitivity.

Thank you!

Below, I offer a few suggestions for improving the manuscript:

Major:

Could the authors please define social sensitivity in the Introduction?

Thank you, we agree that elaborating on this crucial concept is important for an understanding of the scope of this manuscript. We now contrast sensitivity to social information with sensitivity to rewards more clearly and elaborate, also in response to reviewer 3:

Adolescents' behaviours, including their propensity to take risks, are strongly influenced by other people (Blakemore & Mills, 2014; Crone & Dahl, 2012; Steinberg, 2008). Some suggest that this susceptibility to social influence is driven by adolescents' higher sensitivity to rewards in social contexts (Shulman et al., 2016), which makes them particularly prone to taking risks (Chein et al., 2011). Adolescents may also have an increased sensitivity to social information, which may be rooted in attention or motivation to belong to a group (Allen et al., 2022). Both, reward sensitivity and social sensitivity impact susceptibility to social information, which refers to behavioural changes resulting from observing social information. However, it is often overlooked that adolescents can also be more susceptible to social influence because they may simply be more uncertain about what to do.

p. 3. "This uncertainty is part of someone's assumptions about the world, prior to seeing social information. If one's prior assumptions about the utility of their decisions are uncertain, they will be more inclined to listen to what others say and update their beliefs about the utility of the decision after seeing social information." Should this be the utility of the social information for guiding their decisions, not the utility of the decision itself? It would also be helpful for utility to be defined earlier in this paragraph.

Thank you for pointing out how hard it was to understand this paragraph. We now write:

We indeed mean "participants' beliefs about the utility of their decision" and not the utility of social information for guiding decisions. Inferring the latter utility of social information for guiding decisions would be a slightly different process, which requires reasoning about others (FeldmanHall & Shenhav, 2019) and in turn requires researchers to model this process of social learning about others or at least the prior beliefs that participants have about how useful following social information will be.

The utility that people attribute to social information for guiding decisions can differ due to various factors, such as participants' beliefs that others possess more accurate information or because it satisfies a need to belong or signals conformity.

We now write:

[...]Uncertainty is a state of incomplete knowledge and is part of someone's beliefs about the world, for instance about the utility that someone expects from their decisions (Schoemaker, 1982). Experience with similar decisions makes this uncertainty smaller (Hertwig, 2012). In Bayesian learning, uncertain beliefs make novel information the more impactful compared to having less uncertain beliefs. Adolescents may often be more uncertain than adults or children, since they have less personal life-experience to draw from than adults (Rosenbaum et al., 2018), but also more agency about what to do than children. Thus, adolescents' greater uncertainty about the utility of risky decisions could make them more likely to adjust their beliefs and choices in response to social information (FeldmanHall & Nassar, 2021; FeldmanHall & Shenhav, 2019; Konovalov et al., 2018; Laland, 2004; Molleman et al., 2020; Morin et al., 2021; Reiter et al., 2021).

Crucially, while a distinction between the utility of social information and the utility of a decision certainly exists, in practice, we can only observe participants' decisions and how decisions are nudged by social information. The beliefs that people have about others, or different motives for following others, are collapsed in these decisions. Our experiment was not explicitly designed to make inferences about this kind of utility that participants attribute to social information. However, we acknowledge that these differences exist; therefore, we added ψ as a free parameter that can scale the importance a participant attributes to social information. This makes explicit that using social information as an evidence integration process, and we are interested in the endpoint of this process, namely, how strongly it shifts some participants' propensity to choose an option. This is not an unusual assumption and has been modelled elsewhere as an "other conferred utility" (Chung et al., 2015), or "value" or "policy shaping" (Najar et al., 2020).

The Introduction should include more background on age differences in uncertainty to justify hypotheses. Are hypothesized age differences specific to external and/or internal uncertainty?

Thanks for pointing out that we did not provide sufficient background on the uncertainty hypotheses. We also refer to our response of reviewer 1, comment 2 where we elaborate more on uncertainty.

Taken together, we have now added a section to the introduction that better clarifies our use of the internal/external uncertainty terminology. From this and the paragraph that follows, our hypotheses become more justified.

In addressing concerns about our introduction of the preregistered hypotheses, which other reviewers and the editor have also voiced, this new section introduces the concept of "tolerance to uncertainty" in adolescent behaviour, highlighting a contradiction between this idea and the results from the literature on social influence in adolescence. In the social influence literature, many researchers agree that greater uncertainty leads to increased social influence. In the adolescent literature, many agree that adolescents are more susceptible to social influence. We then ask what this means when we combine our knowledge about the mechanisms of social influence with reports of adolescents' tolerance for uncertainty. We are grateful for this comment as these edits underscore the relevance of our work within a current debate.

Can the authors preview the competing hypotheses tested by the alternative cognitive models (e.g., model parameters added)?

We did not initially do that because we did not consider the model comparison as a hypothesis test, but rather introduced the model parameters step by step to make the modelling more accessible. Following good practice, we then compared the full model to simpler versions that are specified by the previous steps to demonstrate that the increased model complexity introduced by additional parameters and the cognitive mechanisms attributed to those parameters is indeed warranted.

But we agree again that a brief explanation would have been helpful, which we now added at the beginning of the modelling section:

The model has four components, each of which specifies a simpler model of task behaviour. In model recovery analyses (see Figure S6), we show that each element contributes to the full model's ability to explain the experiment's behaviour. The simplest model assumes that social information has no impact on choices and that only individual differences in participants' reward sensitivity determine differences in their choices. A more complex specification assumes that social information impacts our participants' decisions in an ideally Bayesian fashion, where social information has a more significant impact when people are more uncertain, and where differences in internal uncertainty correspond to differences in susceptibility to social information. Finally, the most complex model assumes that people may also differ in social sensitivity, which makes social information more impactful than it should be from a Bayesian perspective. We introduce these components step by step in the following.

p. 6. "Participants knew nothing about the advisors' identity, only that we tried to find an advisor matched to their risk preference, which we did by finding an advisor who, on average, made 15 more risky decisions (20%) than the participants in the first half of the experiment."

- What were participants told about the advisor's role and choices in the experiment? Are there post-task manipulation checks to test whether the participants believed the advisor was real or a credible source of information during decision-making? Did participants believe that they were observing the same advisor during the decision from description and decision from experience conditions?

We did not make a post-task manipulation check. However, the question relates to a similar concern raised by the other reviewers, namely, how we ensured that the youngest participants understood the task. We elaborate on this now in the methods section, where we write:

To ensure that even the youngest participants understood the task, an experimenter read out detailed instructions, asked for a comprehension check, and moderated a short practice round before participants started the task. Further there were three built in comprehension check in the instructions, that participants needed to pass before being able to proceed with the experiment (see supplementary material for screenshots of the instructions).

Additionally, we include screenshots of our experiment in the supplementary material, which also illustrates a comprehension check to ensure that all participants understood the experiment

- I commend the authors' effort to match the advisor's choices to participant's initial choices. However, an advisor making 20% more risky choices than the participant seems less like a close match and more like a notably higher level of risky decision-making to observe. Can the authors clarify why the 20% more risky choices threshold was chosen for the "matched" advisor, and why a similar match wasn't made for safer choices as well?

We matched one person to our participants, and this person could have made either riskier or safer choices. A between-participant design to match both would have required a larger participant pool that we did not have the resources for. A within-participant design would have required a longer experiment, which would have been too much of a strain on our youngest participants. Therefore, we had to make a decision. At the time we planned this study, much of the conversation on social influence in adolescent risk-taking was centred on the risky-shift phenomenon, and there was little discussion about safety. Therefore, we designed the task to be most sensitive to age differences in social influence for risky decisions. Of note, a 20% distance was effective in producing a shift towards social information in a different study (Molleman et al., 2020), whereas more distance, possibly because it seems unrealistic to people, reduces social impact (Chung et al., 2015).

Nevertheless, we are grateful for the opportunity to clarify this and have added the a small explanation in the methods section:

[...] we chose this criterion to increase our ability to detect age differences in an increase in risky choices.

- For the Social condition, I do not think seeing how someone else chooses between two jars can be called “advice,” as the latter implies intentional guidance or communication meant to influence another’s decision. I suggest rewording “advice” to something like “observed choices.”

Thanks, we agree that “observed choices” is more accurate. However we thought speaking about “social information” might be even better, because it also allows to talk about “susceptibility to social information”. We updated the text and figures to mention social information and do not speak of advice anymore.

The manuscript currently describes social information as coded ‘-1’ for none, ‘0’ for safe, and ‘1’ for risky advice (p. 8). This coding could be a bit confusing, as it appears to make ‘safe advice’ the reference category rather than ‘no advice,’ which is more commonly used as a neutral baseline. It might help readers if this were clarified, or if the coding were adjusted (e.g., ‘-1’ for safe, ‘0’ for none, and ‘1’ for risky) to make the comparisons and interpretation of effects more intuitive. (You can then use contrast coding to directly compare risky vs. safe advice).

This is very thoughtful advice that we appreciate. However, we had already considered this. If not told otherwise, R automatically uses the smallest number as reference, therefore -1, so no social information becomes the reference category. Also in conjunction with reviewer #3’s similar comments, however, it is clear that we did not explain this well enough. We clarify this now and write:

[...] This way, decisions without social information became a reference category against which the impact of social information favouring the risky or the safe jar could be compared.

Figure 4 caption: Is susceptibility to social influence (difference in prior p and posterior p) the same as social impact? Why use the absolute difference score $|p-p|$ instead of the valenced difference score?

Yes, we refer to this difference as social impact. Thank you for pointing out that the absolute difference may not be the best depiction. We chose this just for the sake of illustration, because we were concerned that the figure would become too cluttered if we showed four lines instead of two. We appreciate the opportunity and encouragement to present our data in a more nuanced way. We paste the novel figure here:

For Figure 4b. Is internal uncertainty ($1/k$) computed separately for decisions from description vs. experience (is that what the blue and gray colors indicate)?

Yes this is what they indicate. We write in the methods section:

We model this internal uncertainty using a Beta distribution, with the lottery's outcome probability p as the mean and a strictly positive rate parameter, which varies depending on whether the decision is based on description or experience.

But this was not clear enough, particularly with respect to the figure, so thanks for pointing to it. In response to the previous comment about Figure 4, we revised the figure caption and took the opportunity to clarify this point. The caption now reads:

Figure 4: Age trends in the social influence model. **a)** Susceptibility to social influence. Difference between the beta distribution mean for p and p' as a measure of social impact (y-axis) by age (x-axis). Black lines and shapes represent decisions from description, and blue lines and shapes represent decisions from experience. Round shapes and solid lines show trials with safe, triangles and dotted lines show trials with risky social information. Positive values indicate that participants were more likely to choose a risky option, and negative values indicate that participants were more likely to choose a safe option after seeing social information. Large shapes show the average in every age bin, the error bars denote the bootstrapped 95% confidence interval, and the lines show a linear fit. **b-e)** Model parameters. Panels depict the population statistics of posterior means of model parameter estimates for each individual (y-axes) by age (x-axes). Boxplots show the interquartile range, whiskers the 95% confidence interval, dots outliers, and the solid line in the box denotes the median of parameter estimates. **b)** Internal uncertainty, which takes separate values for decisions from description (black) and experience (blue). **c)** Social sensitivity, a weight that people put on social information irrespective of their uncertainty, for risky (red) and safe (dark blue) advice. **d)** Reward sensitivity **e)** Temperature or randomness of choices.

To provide a better overview of findings across your sample, it would be helpful to provide in the main manuscript the overview of the (model-free) regression results (i.e., move Table S1 to main manuscript), as

well as descriptives (means, SD, ranges) of key variables of interest from both the behavioral model (e.g., like those shown in Figure S2) and computational model (Figure 4).

Thank you for making the effort to read the supplemental material. We appreciate it, and we happily include a table of the regression results in the main manuscript, which in the revision also includes Bayes factors. The table shows odds ratios and we report the raw regression weights in the text.

However, for descriptive statistics of the variables of interest, we refer to Figures 2 and 4 in the main text, as there were no other variables, and a table for each condition and age bin would be very long and redundant and we would be afraid that this would undermine our efforts to increase the readability of the paper.

To give readers a clearer overview of the findings across your sample, I recommend including the model-free regression results (currently in Table S1) in the main manuscript. It would also be helpful to present descriptive statistics (means, SDs, ranges) for key variables from both the behavioral model (e.g., as in Figure S2) and the computational model (e.g., Figure 4) within the main text.

This comment seems to be a copy-paste error, so it is addressed in the previous response. Please let us know if we are missing something.

p. 19. "In sum, susceptibility to social influence was more pronounced in younger participants than in adults for two reasons." At the end of the Results section, higher internal uncertainty and higher social sensitivity in younger participants was suggested as the two reasons for the observed age-related differences in susceptibility to social influence. Has this mediation pathway (age > internal uncertainty/social sensitivity > susceptibility to social influence) been formally tested? If the mediation model isn't tested or isn't significant, I suggest using less causal language in the Results and Discussion sections.

Thank you for your call for caution. We think the computational modelling justifies the causal language in this case, at least about internal uncertainty and social influence, because this is how the model works and hence the parameters do indeed formally specify the relationship between internal uncertainty and susceptibility to social influence.

I appreciated the interesting, well-written discussion and broader impacts of the study results, especially on potential explanations underlying the observed null quadratic age findings.

Thank you! We think that discussing null results is incredibly important to move science forward.

Minor:

Typo: p. 4. "Finally, adolescents internal uncertainty could be higher, so even if they would do not fear strong negative consequences of a risk they take, they still could be more uncertain than adults if they really want to be part of that group (Reiter et al., 2021)..."

addressed.

p. 13. "(b_EV=4.12, CI = [4.00, 4.24], BF > 100; figure S2a)..." Can the authors please clarify what BF values are?

The reporting of Bayesfactors was not adequate, thank you for pointing this out. Also in response to reviewer #1 we now write explicitly:

We report Bayes factors (BF) as the Savage Dickey density ratio between priors and posterior estimates for the regression weights. In general, Bayes factors obtained this way that are larger than 3 are considered substantial evidence in favour of the alternative hypothesis. Bayesfactors smaller

than 0.3 are considered substantial evidence in favour of the prior (the null hypothesis). Bayesfactors between 0.3 and 3 are considered inconclusive.

I suggest rewording this result to better highlight the significant age effect in social sensitivity, since it currently reads like there were no age differences: "In line with previous research, participants were less sensitive to safe ($b_{age} = -0.15$, CI = [-0.19, -0.12], BF > 100; $b_{age^2} = -0.12$, CI = [-0.16, -0.08], BF > 100 figure 4c, *ψsafe*) and risky ($b_{age} = -0.12$, CI = [-0.16, -0.07], BF > 100; $b_{age^2} = -0.12$, CI = [-0.16, -0.08], BF > 100 figure 4c, *ψrisky*) advice across adolescence" (p. 17).

We agree that this does not read well. With help of this comment and also some comments of reviewer 3, we now edited the results section to be more clear. This result in particular now reads:

In line with previous research (Ciranka & van den Bos, 2019), there were credible negative linear and quadratic age trends for social sensitivity to safe ($b_{age} = -0.15$, CI = [-0.19, -0.12], BF > 100; $b_{age^2} = -0.12$, CI = [-0.16, -0.08], BF > 100 figure 4c,) and to risky ($b_{age} = -0.12$, CI = [-0.16, -0.07], BF > 100; $b_{age^2} = -0.12$, CI = [-0.16, -0.08], BF > 100 figure 4c,) social information.

Reviewer #3 (Remarks to the Author):

I would like to thank the editor and the authors for giving me the opportunity to review this paper, which I found highly interesting. The question raised by this paper - the effect of internal/external uncertainty on susceptibility to social influence during adolescence - is of major importance, as it addresses a significant issue about social influence during adolescence, with far-reaching implications.

Thank you, we appreciate this positive evaluation a lot.

The authors' methodology and analyses are perfectly suited to tackle this critical question, demonstrating a rigorous approach and real expertise in the field. My main reservation about the paper lies in its didactic qualities: the authors should make more efforts to enhance its clarity and accessibility, particularly regarding the hypotheses, the analyses conducted and results. Moreover, I do not entirely agree with the interpretation of one of the key findings of the study regarding the decrease with age in the tendency to follow peer advice. As I will elaborate later, the authors do not consider a central element in this result, whose implications, in my opinion, are nonetheless significant and even support the authors' conclusion regarding the decrease in internal uncertainty with age.

We are convinced that the revisions we made in response to this and the other two reviews made the manuscript clearer and stronger and it helped to clarify our theoretical scope. Thank you for raising the discussion points, which we most of the time fully agree to. The patient's questions and feedback in this review helped us clarify our results and write a more effective research paper.

Even though, this does not undermine the major contribution of this study.

Thank you for communicating the perception that our study has the potential to be a major contribution to the field!

Below, I offer the authors suggestions for further improving their manuscript.

INTRODUCTION

General comments: The authors present a relevant and comprehensive review of the existing literature. I particularly appreciated the emphasis placed on the adaptive value of risk-taking and peer influence during adolescence.

Thank you, we are indeed convinced that this angle of adaptation is essential to understand developmental differences and are delighted when others think so, too.

These aspects, while relatively recent in research discussions, are crucial for understanding developmental processes and decision-making in this age group.

Comment 1:

The authors put forward three explanatory hypotheses to account for the developmental difference in adolescents' social susceptibility. The first suggests greater susceptibility and stronger conformity towards peers. The third posits higher internal uncertainty, making adolescents more easily influenced. As for the second, it requires clarification: I understand that, in a social context, external uncertainty leads to greater internal uncertainty, but in that case, I struggle to see how it differs from the third explanatory hypotheses? Admittedly our motivation behind formulating the hypotheses got lost and we did not express the hypothesis well. We are being more specific now, and also make clear that H4 follows from a result in H1. With these revisions, the differences between H3 and H4 become more apparent.

H3: Adolescents are most influenced by social information.

H4: If H1 is true, then adolescents "uncertainty tolerance", suggests that adolescents will alter their risk-taking behaviour less strongly than adults when external uncertainty increases. In turn, adolescents' susceptibility to social influence should also be less affected by our experimental manipulations of uncertainty.

In the discussion (l. 379-381), the authors conclude: "That is, previous results are compatible with our modelling 378 result that internal uncertainty, that declines with age, is a mechanism behind susceptibility to social influence. 379 This suggests that focusing only on developmental changes in reward sensitivity or the need to belong is 380 insufficient to understand adolescent susceptibility to social influence." I wonder if it would not be more coherent to distinguish in the introduction the following three explanatory hypotheses: 1) greater sensitivity to rewards, 2) the importance of conformity and the sense of belonging, 3) higher internal uncertainty.

Thank you for making it apparent to us that these concepts are not properly explained out of nowhere in our text, although they are so important in the current literature. We took the opportunity to edit our introduction to better reflect these ideas that exist in the literature und write:

Adolescents' behaviours, including their propensity to take risks, are strongly influenced by other people (Blakemore & Mills, 2014; Crone & Dahl, 2012; Steinberg, 2008). Some suggest that this susceptibility to social influence is driven by adolescents' higher sensitivity to rewards in a social context (Shulman et al., 2016), which makes them particularly prone to take risks (Chein et al., 2011). Adolescents may also have an increased sensitivity to social information, which is rooted in attention or motivation to belong to a group (Allen et al., 2022). Both, reward sensitivity and social sensitivity impact susceptibility to social information, which refers to behavioural changes resulting from observing social information. However, it is often overlooked that adolescents can also be more susceptible to social influence because they may simply be more uncertain about what to do.

In response to reviewers 1 and 2 we also elaborated more on the uncertainty hypothesis, so taken together now our introduction and discussion are more consistent with one another and also provides a better overview of the literature.

Comment 2:

Some concepts need further clarification. For instance, while the authors clearly differentiate the terms "social sensitivity" and "susceptibility to social influence" in the introduction, a conceptual definition of these two terms seems essential to help readers fully grasp the distinction. Providing clear explanations would strengthen the paper's accessibility and ensure a better understanding of how these concepts are applied within the study.

We agree that it was unclear what is meant with the distinction between susceptibility to social influence and social sensitivity. We attempted to make this more explicit in the paragraph quoted above.

HYPOTHESES

General comments:

The hypotheses were preregistered, which is highly commendable. However, I find it unfortunate that the hypotheses are much more clearly detailed on OSF than in the article itself. In striving for conciseness, the authors do not provide readers with enough detail to fully understand the hypotheses.

Thank you for this thorough review and looking into OSF and also for recognising that we indeed tried to be concise. We agree that we have become too brief now. We have now added more detail in reporting the preregistered hypotheses in the main text. Furthermore, also in response to reviewer #2 and an earlier concern about lacking clarity regarding hypothesis four raised in this review, we elaborate more on the theoretical background of our hypotheses.

Comment 1: Regarding the first hypothesis (more risky choices among adolescents in decision-from-experience rather than decision-from-description due to greater tolerance for uncertainty), I would like to share a reflection concerning the results of previous studies (e.g., work by Tymula, Blankenstein, etc.). I wonder to what extent the increase in risky choices in conditions comparing a safe option to an ambiguous gamble might be driven by greater sensitivity to rewards rather than a greater tolerance for uncertainty per se. Indeed, the ambiguous gamble offers a much more attractive reward than the safe option. I am fully aware that the same applies to the non-ambiguous gamble and that reward sensitivity should also lead adolescents to prefer the non-ambiguous gamble.

However, I see a fundamental difference here: since probabilities are not available (or only partially) in the ambiguous condition, it is likely that the remaining information—namely, the amount associated with each option—becomes more salient, leading adolescents to choose the ambiguous gamble more frequently given greater reward sensibility. This might explain why adolescents appear as ambiguity averse as adults in tasks inspired by Ellsberg's urns, which directly compare risky and ambiguous options with equivalent associated rewards (see Osmont et al., 2022). I would be curious to hear your thoughts on this issue.

This reflection raises an interesting point about the potential influence of reward sensitivity on adolescents' decision-making in ambiguous situations. When probabilities are unknown, the reward amount may become more prominent in the decision-making process. This could indeed look like a preference for ambiguous gambles, but developmental differences are actually about the reward.

We think that there is one crucial difference in the interpretation of behaviour in the Ellsberg task and tasks with ambiguous gambles inspired by the more recent work initially put forth in (Levy et al., 2010)

In the Ellsberg paradox, conclusions about ambiguity aversion are drawn based on the paradoxical and indicative behaviour of the participants. In a typical scenario, participants are presented with two urns: one containing a known ratio of coloured balls (e.g., 50 red and 50 black) and another containing an unknown ratio. Most people prefer Urn with the known probabilities. The paradox becomes evident when participants are then asked to choose between betting on drawing a black ball from either urn. Again, most prefer Urn A, even though their choices imply contradictory beliefs about the likelihood of drawing red or black from Urn B.

The frequency of such paradoxical choices can be interpreted as “ambiguity aversion” because rewards in these urns are the same, so reward sensitivity should not matter as much, and we can really make an inference about people's beliefs about probabilities.

Now, the ambiguous gambles, in combination with the modelling that was already proposed in Levy et al (2010) are an extension of this, but because the task has so many parameters (value and probability combinations), modelling is required to see it. The equation in Levy et al (2010) and Tymula et al (2012) models ambiguity aversion as a parametric version of the optimism of participants using a parameter β .

$$(p - \beta * \frac{A}{2}) * V^\rho$$

Here, modelling optimism directly changes the probability part, *not the value part*, of the equation. Reward is denoted by V , which, like in our manuscript, is raised to the power of ρ , which directly models interindividual

differences in reward sensitivity. Therefore, using computational modelling, there is a way to reduce the confound of reward sensitivity in the ambiguous lottery task. Studies that do not use this modelling approach potentially have a harder time demonstrating genuine ambiguity aversion since indeed, there may be a confound with reward sensitivity.

That said, we still agree with the central aspect of that reflection, namely that the varying reward may introduce additional reasons for observing developmental differences. While we also do not find an adolescent specific reward sensitivity in the experimental task that we submitted in this manuscript, we acknowledge that through the nature of the tasks that participants' decisions may have more to do with the reward in the ambiguity studies and our experiment than in the Ellsberg paradigm.

We however believe we can claim the differential effects of uncertainty and reward sensitivity by using computational modelling with, and this is essential and shown in the supplement, identifiable (Wilson & Collins, 2019) parameters.

Comment 2: Regarding the second hypothesis (greater sensitivity to peer opinions in decision from experience compared to decision from description), the authors should support their hypothesis by referencing the article by Osmont et al. (2021). This study shows that adolescents refer to the prior choices of their peers in a risk-taking task (BART) only in the uninformed condition (corresponding to decision from experience) and not in the informed condition. Furthermore, this paper demonstrates a greater influence of produced peer choices compared to risky peer influence. The authors could therefore refer to it in several parts of the introduction (e.g., paragraph line 49).

Osmont, A., Camarda, A., Habib M., & Cassotti, M. (2021). Peers' choices influence adolescent risk-taking especially when explicit risk information is lacking. *Journal of Research on Adolescence*, 31 (2), 402-416. <https://doi.org/10.1111/jora.12611>

Thanks for pointing us to this paper; it indeed is quite relevant to our work here. We now cite it in the introduction, where we mention this hypothesis, and also in our discussion, where we discuss that more social influence occurs when there is more internal uncertainty, which is particularly the case in younger participants. We elaborate on this discussion point below in this response, as this review encouraged us to specifically contextualize the finding that there seems to be a developmental asymmetry between safe and risky social information.

Comment 3: Hypothesis 4 needs to be clarified. Referring to the preregistration on OSF, I understand that adolescents, being more tolerant of uncertainty, would exhibit less effect of uncertainty level on their sensitivity to peer advice: they would be sensitive to peer advice in both experience-based and description-based decisions, while adults would be more sensitive in experience-based scenarios. However, following this reasoning, I wonder: if external uncertainty leads to increased internal uncertainty, and internal uncertainty in turn enhances susceptibility to peer advice, then wouldn't adolescents, being more tolerant of uncertainty, experience less internal uncertainty? Consequently, wouldn't this make them less susceptible to peer advice compared to adults? This appears somewhat contradictory with previous works, and I would be curious to hear your perspective on how this interplay is reconciled within the framework of the hypothesis.

In response to this comment and to reviewer 2's similar concerns, we added a brief section on "intolerance to uncertainty" to our introduction, which clarifies how we interpret notions of intolerance to ambiguity. We view it as a description of behaviour in particular environments, rather than a trait-like phenomenon, which also exists but, in our reading, refers to a more general aversion to uncertainty about life's outcomes (Jach et al., 2022). The behavior, that was labeled uncertainty tolerance in adolescents, refers to a higher propensity to make risky decisions in environments with higher uncertainty. It thus describes an age difference in behaviour under higher external uncertainty. Internal uncertainty is separate from that related to social influence; therefore, our (disconfirmed) prediction in hypothesis four is that changes in external uncertainty

lead to fewer differences in susceptibility in social influence in adolescents, because it is their internal uncertainty that drives it.

In terms of mechanisms, being more tolerant to uncertainty could mean that people might be less motivated to reduce their internal uncertainty, thus also stay internally more uncertain and remain more influencable. We think with the addition of the novel section in our introduction, the issue regarding how to interpret statements of “tolerance to uncertainty” becomes clearer.

METHOD

General comments: The methodology is clearly presented and aligns well with the objectives. I only have a few clarifications to highlight and some of the choices made in the design might need additional justification for the sake of transparency.

Comment 1: There are fewer girls than boys in the study. Is this distribution consistent across different age groups? If not, it should be addressed as a limitation. Figure S1 perfectly reports the distribution of age and stages of puberty. I suggest a similar type of figure illustrating the gender distribution by age in the supplementary materials.

Thanks for pointing us to this omission. We strived for an equal gender distribution; however, because the participants were hard to recruit, we could not have a gender balance in every age bin. We however agree that it is information that should be available, thus we updated the figure in the supplement that illustrates the gender distribution in panels as well. We paste the new figure here:

1

Comment 2: In the decision from experience condition, participants learn by observing 9 samples. Why exactly 9 samples were chosen? It would be interesting to know if this number was determined based on pilot studies or theoretical considerations.

¹ Of note, we speculate that there are few “post” adolescent people, particular among our female participants is cultural rather than hormonal because people are asked sensitive questions about their bodies in this questionnaire.

It was the median for adolescents in van den Bos & Hertwig 2017. Thank you for the opportunity to clarify this, We now write:

[...], which was the median number of samples taken by participants under 18 in a previous study (van den Bos & Hertwig, 2017)

As for the outcomes of the samples, clarification is needed. Are the results pre-determined (fixed) to ensure consistency across participants, or are they randomly generated? Providing this information would enhance the transparency and reproducibility of the study.

Yes, we predefined the samples that participants saw on the screen. We did this by ensuring the most representative sample for the underlying probability, meaning that the sample had a mean that was as close as possible to the generative probability value. We now clarify this in the methods section:

We predetermined the samples so that for each condition every participant saw the same number of red and blue marbles. The samples were chosen to be as representative of the generative probability as possible.

Comment 3: Regarding the advisors' choices, the procedure lacks clarity. The justification for presenting the advisor as having a similar risk-taking level to the participant is only mentioned in the discussion. Are the advisor's advices based on the actual responses of a participant who previously completed the task? And why the advisor made 15 more risky decision in the first half of the experience ? It would be helpful to specify exactly how this concept of "matched with their risk preference" is explained to the participants. I wonder how such an abstract notion is perceived and understood by an adolescent of 10 years.

This is not entirely correct; we also mentioned it in the Methods section previously. However, this comment, along with the related comments from the other reviewers, makes us aware that we have not been clear enough. In response to this and the related comments of the other reviewers, we have edited the methods section and now dedicate a subsection to our manipulation of social information.

Experimental manipulation of social information. In the half of the trials, participants saw social information about how someone else chose previously (figure 1 a and b, bottom). Participants knew nothing about the identity of the person whose decisions we showed, only that we tried to find someone matched to their risk preference, which we did by finding an advisor who, on average, made 15 more risky decisions (20%) than the participants in the first half of the experiment. We chose this criterion to increase our ability to detect age differences in an increase in risky choices, but still show conceivably human choices. When it was not possible to find someone who made 15 more risky decisions not being possible, we relaxed this criterion in integer steps until we did.

However, we want to note that the matching to our participants risk-preference doesn't have to be explicitly understood by the participants, we simply did this as an experimental manipulation to make sure that the responses of the others are different enough from the participants' responses, and that it at the same time conceivable that a real person made these choices.

RESULT

General comments: My main concerns refer to the didactic dimension of the presentation of results. Even though I am not an expert in the statistical models used in this study, understanding the results required considerable effort on my part. I found myself having to go back and forth between the article's text and the supplementary material figures, which I occasionally found inconsistent.

This point is gratefully taken and we agree to the didactic issues which the reviews brought to our attention. We believe the impression of inconsistency mostly emerged because of *our presentation* of the results, not

because the results are actually inconsistent. With the help of the suggestions made here, our results no longer appear inconsistent.

This complexity applied even to simpler interaction analyses. Greater cohesion between the text, figures, and supplementary material would certainly improve accessibility and readability.

Comment 1: l.235: The authors highlight an unexpected effect of the advisor's cautious advice: more risky choices. Could this be explained by conformity toward the advisor, who could be generally perceived as more risk-seeking, even during their cautious advice?

Participants might conform to their general perception of the advisor's risk preference rather than to each piece of advice individually. The reference to the 15 riskier choices made in the first part brings me to this idea, although I may have misunderstood this point in the methodology.

Thanks for sharing this thoughtful observation. Indeed, this could be an explanation that aligns with a preference shift that occurs when predicting the choices of someone with a different preference, as reported by Reiter et al. (2021). We have now included this potential explanation in our discussion, and we will elaborate on this alongside another concern raised below in this review.

Comment 2:

l. 250: The term "experience condition" is indeed potentially confusing, as it more naturally refers to the decision-from-experience condition.

We agree and now consistently write 'decisions-from-experience' when referring to this condition specifically. Thank you.

l. 253: The authors mention an interaction between the linear age term and the level of uncertainty (experience versus description). They state "participants' propensity to choose risky decreased with age when making decisions from experience ". However, what about the decision from description condition? In Figure S2i, the slope appears less pronounced compared to the decision from experience condition, but there is still a noticeable decrease with age.

Thank you for making the effort to check the supplement. We realised, also in the light of the following comments, that throughout we did not do a good job at explaining the interactions that we find, because we did not explain the reference categories of the regression model. We elaborate on the reference categories and interactions more in the methods section now, but also updated our mentions of the interactions to reflect the relative difference between groups and not an absolute trend. We paste our explanations after the response to the next comment.

Comment 3:

l. 255-263 (and after): I am not entirely convinced by the way the authors operationalize susceptibility to peer advice in their results. The authors interpret susceptibility solely based on the safe advice trials (or risky advice). However, I believe that the discussion about susceptibility to the advisor's advice relies on a comparison between the advice condition (safe or risky) and the control condition (none).

For instance, based on Figure S2h, the following questions arise:

- Does the safe/experience condition differ from the None/experience condition?
- Does the risky/experience condition differ from the None/experience condition?
- Does the safe/description condition differ from the None/description condition?
- Does the risky/description condition differ from the None/description condition?

Thanks for pointing us to that we did not provide enough information to interpret our regression results. The issue is the same reference category problem that has been mentioned by reviewer 2 as well. All of the "social" results need to be interpreted in reference to the "no social information" condition. We better clarify

in the methods section how the reference categories are determined and what this means for interpreting the regression weights:

All regressions used generalized linear mixed models to predict each decision using a logit link function to a Bernoulli likelihood. In this model, a risky choice was coded as "1" and a safe choice as "0". For the regressors, decisions based on description were coded as "0" and decisions based on experience were coded as "1". Social information was coded as follows: "-1" for no social information, "0" for safe social information, and "1" for risky social information. This coding makes "no social information" the reference category, allowing us to compare the impact of advice favouring either the risky or safe option. Any regression term involving social information (whether it was risky or safe) or its interactions with another factor evaluates the difference between choices made without social information and those made with either risky or safe social information.

Comment 4:

l. 264-274: Concerning H3, the results are somewhat difficult to follow.

Regarding the decision from description condition, it is evident that there is sensitivity to risky advice, which remains consistent across all ages (as observed in the comparison between the "risky" and "none" lines). However, for the decision from experience condition, I don't entirely align with the authors' conclusions. Based on Figures S2k and, more notably, S2i, it appears that younger participants tend to follow cautious advice, resulting in less risk-taking compared to the "none" condition. On the other hand, for older participants, I would have concluded to an unexpected effect of cautious advice—it seems clear that there is an increase in risky choices compared to the control condition.

This observation is correct. We think that through the table we added in this revision, in response to reviewer 2, the regression results become more interpretable. We also elaborate now:

These linear and quadratic effects taken together suggest an initially strong and then plateauing age-related decline of susceptibility to social influence after seeing safe social information, which, judging from visually inspecting the marginal effect predictions (figure S2k), might even be reversed in the oldest participants. The oldest participants seem more likely to choose risky when seeing safe social information, compared to when seeing no social information.

We need to note, however that figure S2i shows an orthogonal squared age and not linear age on the x axis, because for these marginal plots quadratic age, and not linear age, was the predictor. Figure S2i can therefore not be interpreted as a linear age trend. 10 and 26-year-old participants (oldest and youngest) are **both** represented in the leftmost part, and 18-year-old participants are described in the rightmost part of the plot. There is an additional issue with interpreting orthogonal linear and quadratic age polynomials in isolation, since in effect they add up. When linear and quadratic age trends are present, this results in a steep change of the function in the beginning, which then plateaus, which reflects the overall pattern in our data. We are afraid, however, that elaborating on this statistical peculiarity at length is beyond the scope of this article.

Comment 5:

l. 282-285: Concerning H4, the results are also difficult to follow : “with increasing age, advice favouring the safe jar was related to more risky decisions when participants made decisions from experience, where external uncertainty was higher ($b_{\text{safe}} \times \text{experience} \times \text{age} = 283.045$ CI = [0.1, 0.8], BF = 52.00; figure S2m), suggesting that younger participants were steered away from choosing risky more strongly than older participants when external uncertainty was high.”

If, with age, risky advice led to more risky choices, the blue curve would be ascending. However, it remains slightly descending. Are the authors rather referring to the quadratic effect of age (f.S2n instead of f.S2m)?

In figure f.S2m we observe that older participants in the decision from experience condition oppose their peers' advice and make more risky choices compared to the control condition.

This is correct, if risky advice would lead to more changes overall, the blue curve would indeed be ascending. Thank you for pointing us to another flawed wording, which does not appropriately reflect the three-way interaction. We note that in the revised version with new analyses this result is not credible anymore. This is the only result that changed based on our novel way to compute Bayes factors. This also speaks for that it was not a strong result in the first place which may explain the difficulty to interpret the result based on the marginal effect plots.

The interaction was small to start with and using a more conservative prior and calculating proper Bayes factors as opposed to evidence ratios, in response to reviewer 1 we get a Bayes factor around 2 which does not surpass the conventional threshold of what is regarded as a substantial effect (which is 3). We now conclude:

This means that there was no developmental difference in the interaction between different degrees of external uncertainty and risky or safe social information.

Which is also more in line with the observation shared in this comment, so thanks for your vigilance!

Comment 6:

l. 286-289: "In sum, behavioural results indicate that older participants choose more cautiously than younger ones when 286 there is high external uncertainty requiring decisions from experience, with no adolescent peak in risky choices. 287 Overall, participants were more susceptible to social influence when advice favoured the safe, but not the risky 288 jar, when external uncertainty was high and required decisions from experience." The authors provide a summary of the main results in connection with their hypotheses, which is indeed necessary. However, I only partially agree with their conclusions. While I align with the authors' first statement, I disagree with the second. It seems to me that exposure to risky advice influences participants' choices in the decision-from-experience situation, regardless of their age, by increasing the number of risky choices. On the other hand, when it comes to cautious advice, younger participants tend to follow this advice, whereas older participants seem to oppose it.

Thanks again for another insightful comment. We agree with the interpretation that young people use safe social information more, which also becomes more evident in the new Figure 4. Although, when modelling and behavioral regression results are taken together we are not (internally) certain that older adults indeed oppose safe advice and how much of this is caused by the structure of the experimental environment, which we think explains our error in summarising here and we get back to this in the discussion and elaborate at a later point in the review. We now write at this point:

Summarising the age trends of our behavioural results indicates that older participants choose more cautiously than younger ones when there is high external uncertainty requiring decisions from experience, with no adolescent peak in risky choices. Overall, younger participants seemed more susceptible to social influence when social information favoured the safe, but not the risky jar, a difference that remained statistically similar when external uncertainty was low or high.

DISCUSSION

General comments: The discussion presented here is well-conducted and effectively highlights what I consider to be the most important result of this study: the decrease in internal uncertainty during adolescence, which

appears to be a central mechanism for sensitivity to social influence. However, several points must be further discussed.

Thanks, below are all good points and we appreciate their thoughtfulness.

Comment 1: I would have liked the authors to further discuss their first hypothesis. While they conclude from their results a decrease in risky choices with age in the decision from experience condition, they do not address the decision from description condition. Even though the slope is steeper for the decision from experience condition (figure S2i and j), it appears a quite similar decrease for the decision from description condition. Therefore, these results do not seem to truly validate H1 or support previous studies. Personally, I am quite convinced that adolescents experience greater uncertainty, but not that they are more tolerant of this uncertainty (as noted in Hypothesis comment 1). Although I agree with Tymula's idea that tolerance to uncertainty holds essential adaptive value for adolescents, I believe that most of them are already highly intolerant of it.

Thanks, we agree. However, we can not say much about the interpretation of Tymula et al, because this paradigm is very different from ours. We focus on "uncertainty" as we operationalise it instead, and now elaborate:

In line with many previous studies (Defoe et al., 2015), the youngest participants decided for the risky option the most often and this behaviour declined linearly with age. Thus, there was no peak in adolescents' propensity to take risks, which would imply a special reward sensitivity of adolescents in our task. On average, all participants made fewer risky decisions when uncertainty in the experiment was higher, there was no adolescent peak in choosing risky under uncertainty. Thus, hypothesis H1 that adolescents would be tolerant to uncertainty and choose risky most often when uncertainty is high was not confirmed. Because of that, hypothesis H4, that because adolescents are uncertainty tolerant, their social susceptibility would not depend as much on external uncertainty could also not be supported.

We elaborate on this further in the revised manuscript.

Comment 2: Once again, the result indicating that older participants decide against the advisor cautious choices in the decision from experience condition seems to be ignored by the authors. In my view, this finding further reinforces the aforementioned conclusion. Younger adolescents are more likely to follow their peers' cautious choices due to greater internal uncertainty, unlike young adults. Thus, internal certainty, which increases with age, appears to be crucial for resisting peer influence and even opposing their advice. This point warrants further discussion.

Thank you for the opportunity to talk about this exciting aspect of the data. There are actually two plausible explanations for the propensity to disregard safe information that are both plausible at first sight but only one is compatible with the developmental pattern we find. We elaborate on this in the discussion, which makes it more sound and scientifically interesting. We point out different aspects now.

We do not copy all paragraphs that contain the respective edits, because they are plenty. Instead we very briefly summarize the two added arguments.

First, as this review brought to our attention, relative to adults; youngest people are still more persuaded by safe social information and the difference in social impact when comparing the youngest and oldest participants in our sample is potentially even stronger for safe social information, which is in line with Osmont et al. 2021 and also other previous work.

Second, that safe social information is less impactful could be because participants undergo a preference shift from observing social information or it could be because they think social information for the safe options is unreasonable. The latter explains adults anti-conformity better than the preference shift, which we, based on the literature, would expect to be stronger in the youngest and not the oldest participants.

Comment 3: l. 409-411: "Since we matched advisors to our participants' choices, the tendency to use risky advice more, which 409 we observe consistently in participants older than 13 but not in younger participants, may reflect an emerging 410 ability to discern advice that feels useful to participants from advice that does not (Moses-Payne et al., 2021)."

Another point should be further discussed. The results of this study also challenge the stereotypical view of adolescents as particularly vulnerable to risky peer influence and therefore more likely to take risks to follow their peers. Here, we note that susceptibility to the advisor's risky influence is not age dependent. Only sensitivity to cautious influence is higher in early adolescence. Thus, adolescents do not follow the advisor's risky advice more often; instead, they are more sensitive to their cautious advice. This point is also supported by the findings of Osmont et al. (2021), suggesting that peers' prior cautious choices have more impact on adolescents' risk-taking decisions than previously risky choices in ambiguous situations. Adolescents follow their peers' risky choices only when these align with the trial's risk level, also indicating discernment in facing peer influence.

We agree that our results challenge the stereotypical view, however we need to note that also risky social information is more impactful in early adolescents compared to adults, which is possibly more visible in the new figure 4 that we added in response to a comment of reviewer #2. The revisions we made in response to the previous point make more clear that safe social information seems to be particularly influential when directly comparing its impact in adults and early adolescents.

We took this comment as another encouragement to elaborate more on the remark that there is no adolescent peak in reward sensitivity.

Comment 4: l. 397-430:

In this paragraph, the authors should reference the comparative ignorance hypothesis:

Fox, C. R., & Tversky, A. (1995). Ambiguity Aversion and Comparative Ignorance. *The Quarterly Journal of Economics*, 110(3), 585-603. <https://doi.org/10.2307/2946693>

After the revisions, this paragraph does not exist anymore, but we think it is an extremely interesting paper, has its place when discussing the main effect of uncertainty where we now write:

On average, all participants made fewer risky decisions when uncertainty in the experiment was higher, potentially demonstrating a case of "comparative ignorance", where people prefer less uncertain options when they can directly compare them to more uncertain options (Fox & Tversky, 1995).

TYPOGRAPHICAL REVISIONS

l. 84-85: The sentence seems incomplete.

l. 32 : "und" instead of "and"

Thank you for pointing us to these errors, we carefully edited the manuscript again and hope that we eliminated all remaining mistakes.

REFERENCES

- Bach, D. R., & Dolan, R. J. (2012). Knowing how much you don't know: a neural organization of uncertainty estimates. *Nature Reviews. Neuroscience*, *13*(8), 572–586. <https://doi.org/10.1038/nrn3289>
- Blankenstein, N. E., Crone, E. A., van den Bos, W., & van Duijvenvoorde, A. C. K. (2016). Dealing With Uncertainty: Testing Risk- and Ambiguity-Attitude Across Adolescence. *Developmental Neuropsychology*, *41*(1-2), 77–92. <https://doi.org/10.1080/87565641.2016.1158265>
- Braams, B. R., Davidow, J. Y., & Somerville, L. H. (2019). Developmental patterns of change in the influence of safe and risky peer choices on risky decision-making. *Developmental Science*, *22*(1), e12717. <https://doi.org/10.1111/desc.12717>
- Chung, D., Christopoulos, G. I., King-Casas, B., Ball, S. B., & Chiu, P. H. (2015). Social signals of safety and risk confer utility and have asymmetric effects on observers' choices. *Nature Neuroscience*, *18*(6), 912–916. <https://doi.org/10.1038/nn.4022>
- FeldmanHall, O., & Shenhav, A. (2019). Resolving uncertainty in a social world. *Nature Human Behaviour*, *3*(5), 426–435. <https://doi.org/10.1038/s41562-019-0590-x>
- Giron, A. P., Ciranka, S., Schulz, E., van den Bos, W., Ruggeri, A., Meder, B., & Wu, C. M. (2023). Developmental changes in exploration resemble stochastic optimization. *Nature Human Behaviour*. <https://doi.org/10.1038/s41562-023-01662-1>
- Jach, H. K., DeYoung, C. G., & Smillie, L. D. (2022). Why do people seek information? The role of personality traits and situation perception. *Journal of Experimental Psychology. General*, *151*(4), 934–959. <https://doi.org/10.1037/xge0001109>
- Juechems, K., Balaguer, J., Spitzer, B., & Summerfield, C. (2021). Optimal utility and probability functions for agents with finite computational precision. *Proceedings of the National Academy of Sciences of the United States of America*, *118*(2). <https://doi.org/10.1073/pnas.2002232118>
- Kay, J. (2020). *Radical Uncertainty*. https://www.suerf.org/wp-content/uploads/2023/12/f_09504c80bbd8bb4b75f97df27a9feaa3_16053_suerf.pdf
- Knight, F. H. (1921). *Risk, uncertainty and profit: Vol. XXXI*. Houghton Mifflin Company.

<https://archive.org/details/riskuncertainty00knigrich>

Kozyreva, A., Pleskac, T. J., Pachur, T., & Hertwig, R. (2019). Interpreting uncertainty: A brief history of not knowing. In *Taming Uncertainty* (pp. 343–362). The MIT Press.

<https://doi.org/10.7551/mitpress/11114.003.0026>

Levy, I., Snell, J., Nelson, A. J., Rustichini, A., & Glimcher, P. W. (2010). Neural representation of subjective value under risk and ambiguity. *Journal of Neurophysiology*, *103*(2), 1036–1047.

<https://doi.org/10.1152/jn.00853.2009>

Meder, B., Wu, C. M., Schulz, E., & Ruggeri, A. (2021). Development of directed and random exploration in children. *Developmental Science*, *24*(4), e13095. <https://doi.org/10.1111/desc.13095>

Molleman, L., Tump, A. N., Gradassi, A., Herzog, S., Jayles, B., Kurvers, R. H. J. M., & van den Bos, W. (2020). Strategies for integrating disparate social information. *Proceedings. Biological Sciences / The Royal Society*, *287*(1939), 20202413. <https://doi.org/10.1098/rspb.2020.2413>

Moutoussis, M., Dolan, R. J., & Dayan, P. (2016). How People Use Social Information to Find out What to Want in the Paradigmatic Case of Inter-temporal Preferences. *PLoS Computational Biology*, *12*(7), e1004965. <https://doi.org/10.1371/journal.pcbi.1004965>

Najar, A., Bonnet, E., Bahrami, B., & Palminteri, S. (2020). The actions of others act as a pseudo-reward to drive imitation in the context of social reinforcement learning. *PLoS Biology*, *18*(12), e3001028. <https://doi.org/10.1371/journal.pbio.3001028>

Palminteri, S., Kilford, E. J., Coricelli, G., & Blakemore, S.-J. (2016). The Computational Development of Reinforcement Learning during Adolescence. *PLoS Computational Biology*, *12*(6), e1004953. <https://doi.org/10.1371/journal.pcbi.1004953>

Reiter, A. M. F., Moutoussis, M., Vanes, L., Kievit, R., Bullmore, E. T., Goodyer, I. M., Fonagy, P., Jones, P. B., NSPN Consortium, & Dolan, R. J. (2021). Preference uncertainty accounts for developmental effects on susceptibility to peer influence in adolescence. *Nature Communications*, *12*(1), 1–13. <https://doi.org/10.1038/s41467-021-23671-2>

Schulz, E., Wu, C. M., Ruggeri, A., & Meder, B. (2019). Searching for Rewards Like a Child Means Less Generalization and More Directed Exploration. *Psychological Science*, *30*(11), 1561–1572. <https://doi.org/10.1177/0956797619863663>

Tversky, A., & Kahneman, D. (1992). Advances in prospect theory: Cumulative representation of uncertainty.

Journal of Risk and Uncertainty, 5(4), 297–323. <https://doi.org/10.1007/BF00122574>

Volz, K. G., & Gigerenzer, G. (2012). Cognitive Processes in Decisions Under Risk are not the Same as in

Decisions Under Uncertainty. *Frontiers in Neuroscience*, 6, 105.

<https://doi.org/10.3389/fnins.2012.00105>

Wilson, R. C., & Collins, A. G. E. (2019). Ten simple rules for the computational modeling of behavioral data.

eLife, 8, e49547. <https://doi.org/10.7554/eLife.49547>

Reviewer #2 (Remarks to the Author):

I appreciate the authors' extensive revisions and responses to reviewer comments and believe the current paper will make an exciting contribution to the adolescent social influence literature. The comprehensive revisions, especially in clarifying the definitions of key terms and hypotheses and variables' coding and interpretation, address many of the previously raised concerns.

Thanks a lot for this positive evaluation.

Two minor suggestions remain to improve the clarity of the Introduction:

Knight 1921's definitions of epistemic and aleatoric uncertainty were a valuable addition to the theoretical foundation and chosen taxonomy presented in the Introduction. I think the authors' reviewer response noting that some types of uncertainty are reduceable through learning than others, whereas others are not is conceptually important to include in the main manuscript. To that end, I suggest a minor edit to the following sentence (p. 3, Introduction): "Thus, there is internal uncertainty in someone's beliefs about the utility of a choice that can be reduced through learning and experience and there is external uncertainty that is a feature of the unpredictability of the environment itself that cannot be reduced (Knight, 1921)."

We now use this sentence suggested here and agree it communicates the nuance of different uncertainty types better.

In general, the revised hypotheses are easier to understand, but I think the reference groups for H3 and H4 remain unclear and should be made more explicit. For H3, adolescents are most influence by social information compared to children and adults? Compared to no social information? For H4, adolescents' susceptibility to risky social influence should be less affected by our experimental manipulations of uncertainty compared to children and adults?

When talking about "adolescent peaks" or similar wordings we always meant with reference to both, adults and children. We now clarified this.

Reviewer #3 (Remarks to the Author):

I would like to sincerely thank the authors for the quality and precision of their responses to each of the comments raised by the three reviewers. Their thorough work has helped clarify key points in the article, made the results and conclusions more understandable, and highlighted the valuable contributions of this study. I am therefore in favor of the publication of this article.

Thank you, we enjoyed the discussion and thought the reviews made our paper better.

My only remaining concern is that Figure S2, which is referenced repeatedly in the results section, has been placed in the supplementary materials. It serves as an essential tool for understanding the findings and, in my view, should be included in the main article (at least in part) to avoid requiring readers to constantly navigate between the paper and the supplementary material. Once again, thank you for inviting me to participate in this review, which has been stimulating and enriching.

Thanks for the encouragement to include the figure in the main text. We also think it is helpful, but were afraid that it deters too much from the main argument, namely that it is important to consider internal uncertainty. To stay closer to the main contribution of the paper, we now we omit the marginal effects for working memory and fluid intelligence, as we also did not really reference them in the text and included a prettier figure in the main manuscript. There is a caveat however, that also came apparent in this discussion. We were afraid that the marginal effects of the quadratic age regressor might be deceptive, because first, it is not intuitive what the x-axis represents and second separating linear and quadratic age in this visualization suggests that we could take apart age into separate components, which is not true, albeit our preregistration reads as if this would be what a quadratic polynomial does. We came to understand these polynomials as a way to model non-linear differences between age groups. In order to see what the regression model really represents it is more helpful to visualize its predictions instead, which we now added to this marginal effects plot as well. We also address this caveat in the figure caption and hope this is a good compromise. We paste it below:

Figure 4: Regression results (n=166 participants). **a)** Shows regression model predicted age trends in risky choice when there was no social information (green), social information favouring the safe (orange) or risky (purple) marble. Each line refers to predictions of one sample from the posteriors of the regression model. **b-m)** Denote marginal effects and depict how the probability to choose risky (y-axis) differs with changes in the independent variables (x-axes), holding other variables constant. Two-way interactions are illustrated with colours. Interactions with age and social information are represented with green, orange and purple for none, safe and risky social information (h & l). External uncertainty is represented with blue and black when it was high and low, respectively. Three-way interactions of age, uncertainty and social information (panels i and m) always show uncertainty conditions in panels and social information in colour. The second row shows linear and the third row shows quadratic polynomials of our participant's age. The quadratic marginal effect represents both, the youngest and the oldest participants at the left side of the x axis, whereas the rightmost side of the x axis represents 18year olds. It is important to note that these rows should not be interpreted in isolation. The polynomials are needed to model non-linear age differences and are combined in the predictions shown in a). Errorbars (c-d) and ribbons (a & f-m) show the effects that lie within the 95% credible interval of the regression weight.